# Inhibition of *Cpeb3* ribozyme elevates CPEB3 protein expression and polyadenylation of its target mRNAs and enhances object location memory

Claire C Chen[1], Joseph Han[2], Carlene A Chinn[2], Jacob S Rounds[2], Xiang Li[2†], Mehran Nikan[3], Marie Myszka[4], Liqi Tong[5], Luiz FM Passalacqua[1], Timothy Bredy[2†], Marcelo A Wood[2]*, Andrej Luptak[1,4,6]*

[1]Department of Pharmaceutical Sciences, University of California, Irvine, Irvine, United States; [2]Department of Neurobiology and Behavior, Center for the Neurobiology of Learning and Memory, University of California, Irvine, Irvine, United States; [3]Ionis Pharmaceuticals, Carlsbad, United States; [4]Department of Chemistry, University of California, Irvine, Irvine, United States; [5]Institute for Memory Impairments and Neurological Disorders, University of California, Irvine, Irvine, United States; [6]Department of Molecular Biology and Biochemistry, University of California, Irvine, Irvine, United States

*For correspondence:
mwood@uci.edu (MAW);
aluptak@uci.edu (AL)

Present address: †Cognitive Neuroepigenetics Laboratory, Queensland Brain Institute, University of Queensland, Brisbane, Australia

**Abstract** A self-cleaving ribozyme that maps to an intron of the cytoplasmic polyadenylation element-binding protein 3 (*Cpeb3*) gene is thought to play a role in human episodic memory, but the underlying mechanisms mediating this effect are not known. We tested the activity of the murine sequence and found that the ribozyme's self-scission half-life matches the time it takes an RNA polymerase to reach the immediate downstream exon, suggesting that the ribozyme-dependent intron cleavage is tuned to co-transcriptional splicing of the *Cpeb3* mRNA. Our studies also reveal that the murine ribozyme modulates maturation of its harboring mRNA in both cultured cortical neurons and the hippocampus: inhibition of the ribozyme using an antisense oligonucleotide leads to increased CPEB3 protein expression, which enhances polyadenylation and translation of localized plasticity-related target mRNAs, and subsequently strengthens hippocampal-dependent long-term memory. These findings reveal a previously unknown role for self-cleaving ribozyme activity in regulating experience-induced co-transcriptional and local translational processes required for learning and memory.

## Editor's evaluation

In this manuscript the authors describe the expression and regulatory function of a self-cleaving ribozyme in the *Cpeb3* gene. This is an important study because although self-cleaving ribozymes have been identified in the genome, the functions of these RNA enzymes for molecular control for the genes that harbor them is mostly unknown. The manuscript provides compelling data for the molecular function of the ribozyme in gene expression regulation and solid evidence of its role in hippocampal learning. The study will be of interest to neurobiologists who study gene regulatory mechanism.

**eLife digest** Stored within DNA are the instructions cells need to make proteins. In order for proteins to get made, the region of DNA that codes for the desired protein (known as the gene) must first be copied into a molecule called messenger RNA (or mRNA for short). Once transcribed, the mRNA undergoes further modifications, including removing redundant segments known as introns. It then travels to molecular machines that translate its genetic sequence into the building blocks of the protein.

Following transcription, some RNAs can fold into catalytic segments known as self-cleaving ribozymes which promote the scission of their own genetic sequence. One such ribozyme resides in the intron of a gene for CPEB3, a protein which adds a poly(A) tail to various mRNAs, including some involved in learning and memory. Although this ribozyme is found in most mammals, its biological role is poorly understood.

Previous studies suggested that the ribozyme cleaves itself at the same time as the mRNA for CPEB3 is transcribed. This led Chen et al. to hypothesize that the rate at which these two events occur impacts the amount of CPEB3 produced, resulting in changes in memory and learning. If the ribozyme cleaves quickly, the intron is disrupted and may not be properly removed, leading to less CPEB3 being made. However, if the ribozyme is inhibited, the intron remains intact and is efficiently excised, resulting in higher levels of CPEB3 protein.

To test how the ribozyme impacts CPEB3 production, Chen et al. inhibited the enzyme from cutting itself with antisense oligonucleotides (ASOs). The ASOs were applied to in vitro transcription systems, neurons cultured in the laboratory and the brains of living mice in an area called the hippocampus.

The in vitro and cell culture experiments led to higher levels of CPEB3 protein and the addition of more poly(A) tails to mRNAs involved in neuron communication. Injection of the ASOs into the brains of mice had the same effect, and also improved their memory and learning.

The findings of Chen et al. show a new mechanism for controlling protein production, and suggest that ASOs could be used to increase the levels of CPEB3 and modulate neuronal activity. This is the first time a biological role for a self-cleaving ribozyme in mammals has been identified, and the approach used could be applied to investigate the function of two other self-cleaving ribozymes located in introns in humans.

## Introduction

Cytoplasmic polyadenylation element-binding proteins (CPEBs) are RNA-binding proteins that modulate polyadenylation-induced mRNA translation, which is essential for the persistence of memory (*Huang et al., 2003*). CPEBs have been found in several invertebrate and vertebrate genomes, and four *Cpeb* genes (*Cpeb1–4*) have been identified in mammals (*Si et al., 2003*; *Theis et al., 2003*; *Richter, 2007*; *Merkel et al., 2013*; *Afroz et al., 2014*). All CPEB proteins have two RNA recognition domains (RRM motifs) and a ZZ-type zinc finger domain in the C-terminal region, but they differ in their N-terminal domains (*Hake and Richter, 1994*; *Huang et al., 2006*; *Ivshina et al., 2014*). *Aplysia* CPEB (ApCPEB), *Drosophila* Orb2, and mouse CPEB3 have two distinct functional conformations that correspond to soluble monomers and amyloidogenic oligomers, and have been implicated in the maintenance of long-term facilitation (LTF) in *Aplysia* and long-term memory in both *Drosophila* and mice (*Miniaci et al., 2008*; *Si et al., 2010*; *Majumdar et al., 2012*; *Fioriti et al., 2015*; *Hervás et al., 2016*; *Rayman and Kandel, 2017*; *Hervas et al., 2020*). In *Drosophila*, inhibition of amyloid-like oligomerization of Orb2 impairs the persistence of long-lasting memory, and deletion of the prion-like domain of Orb2 disrupts long-term courtship memory (*Keleman et al., 2007*; *Hervás et al., 2016*). The aggregated form of CPEB3, which is inhibited by SUMOylation, can mediate target mRNA translation at activated synapses (*Drisaldi et al., 2015*).

Following synaptic stimulation, CPEB3 interacts with the actin cytoskeleton, with a positive feedback loop of CPEB3/actin regulating remodeling of synaptic structure and connections (*Stephan et al., 2015*; *Gu et al., 2020*). Studies of CPEB3 in memory formation revealed that local protein synthesis and long-term memory storage are regulated by the prion-like CPEB3 aggregates, which are thought to strengthen synaptic plasticity in the hippocampus. While *Cpeb3* conditional knockout mice display impairments in memory consolidation, object placement recognition, and long-term memory

maintenance (**Fioriti et al., 2015**), global *Cpeb3* knockout (*Cpeb3*-KO) mice exhibit (i) enhanced spatial memory consolidation in the Morris water maze (MWM), (ii) elevated short-term fear memory in a contextual fear conditioning task, and (iii) improved long-term memory in a spatial memory task (water maze) (**Chao et al., 2013**). Moreover, dysregulation of translation of plasticity-associated proteins and post-traumatic stress disorder-like behavior after traumatic exposure is observed in *Cpeb3*-KO mice (**Lu et al., 2021**).

In addition to encoding the CPEB3 protein, the mammalian *Cpeb3* gene also encodes a functionally conserved self-cleaving ribozyme that maps to the second intron (**Salehi-Ashtiani et al., 2006**; **Webb and Lupták, 2011**; **Bendixsen et al., 2021**; *Figure 1A*). Several mammalian ribozymes have been identified (**Sharmeen et al., 1988**; **Wu et al., 1989**; **Salehi-Ashtiani et al., 2006**; **Martick et al., 2008**; **de la Peña and García-Robles, 2010**; **Perreault et al., 2011**; **Hernandez et al., 2020**; **Chen et al., 2021**), including the highly active sequence in the *Cpeb3* gene. The *Cpeb3* ribozyme belongs to hepatitis delta virus (HDV)-like ribozymes, which are self-cleaving RNAs widespread among genomes of eukaryotes, bacteria, and viruses (**Webb et al., 2009**; **Eickbush and Eickbush, 2010**; **Ruminski et al., 2011**; **Sánchez-Luque et al., 2011**; **Weinberg et al., 2015**). The biological roles of these ribozymes vary widely and include processing rolling-circle transcripts during HDV replication (**Sharmeen et al., 1988**; **Wu et al., 1989**), 5'-cleavage of retrotransposons (**Eickbush and Eickbush, 2010**; **Ruminski et al., 2011**; **Sánchez-Luque et al., 2011**), and in one bacterial example, the HDV-like ribozyme may mediate metabolite-dependent regulation of gene expression (**Passalacqua et al., 2017**). Furthermore, the genomic locations of these catalytic RNAs suggest that they are involved in many other biological processes. Recent analysis suggests that *Cpeb3* ribozymes have had a role in mammals for over 100 million years, although their biological function remains unknown (**Bendixsen et al., 2021**). In humans, a single-nucleotide polymorphism (SNP) at the ribozyme cleavage site leads to a threefold higher rate of in vitro self-scission, which correlates with poorer performance in an episodic memory task (**Salehi-Ashtiani et al., 2006**; **Vogler et al., 2009**) and suggests that the ribozyme activity may play a role in memory formation.

While the CPEB3 protein is well established as a modulator of memory formation and learning, the molecular and physiological functions of the intronic *Cpeb3* ribozyme have not been tested. Using synthetic ribozymes placed within introns of mammalian genes, previous work showed that splicing of the surrounding exons is sensitive to the continuity of the intron: fast ribozymes caused efficient self-scission of the intron, leading to unspliced mRNA and lower protein expression. In contrast, slow ribozymes had no effect on mRNA splicing and subsequent protein expression (**Fong et al., 2009**). Based on this observation, we tested the hypothesis that inhibition of the *Cpeb3* ribozyme co-transcriptional self-scission will promote *Cpeb3* mRNA splicing (*Figure 1A*) and increase the expression of full-length mRNA and CPEB3 protein, leading to polyadenylation of its target mRNAs and enhancement in the consolidation of hippocampal-dependent memory.

## Results

### Antisense oligonucleotides (ASOs) inhibit *Cpeb3* ribozyme self-scission

To determine whether the *Cpeb3* ribozyme activity modulates expression of the CPEB3 protein by disrupting co-transcriptional splicing of the *Cpeb3* mRNA, we started by measuring the co-transcriptional self-scission of the murine variant of the ribozyme in vitro and determined the half-life ($t_{1/2}$) to be ~2–3 min (*Figure 1B* and *Table 1*). This rate of self-scission is similar to that measured previously for chimp and fast-reacting human variants of the ribozyme (**Chadalavada et al., 2010**). Because the distance from the ribozyme cleavage site to the third exon in the *Cpeb3* gene is 9931 nucleotides (*Figure 1A*) and the RNA polymerase II (RNAPII) transcription rate of long mammalian genes is estimated to be ~3.5–4.1 knt/min (**Singh and Padgett, 2009**), RNAPII should require about 2.5–3 min to travel from the ribozyme to the third exon. The nascent ribozyme thus self-cleaves in about the same time as it takes the RNAPII to synthesize the remaining part of the intron and the next exon, at which point the splicing machinery is expected to mark the intron–exon junction. This observation suggests that the ribozyme activity is tuned to the co-transcriptional processing of the *Cpeb3* pre-mRNA: a significantly faster rate of self-scission would lead to a high fraction of cleaved, unspliced pre-mRNAs, whereas slow self-cleavage rate would have no effect on the *Cpeb3* pre-mRNA splicing.

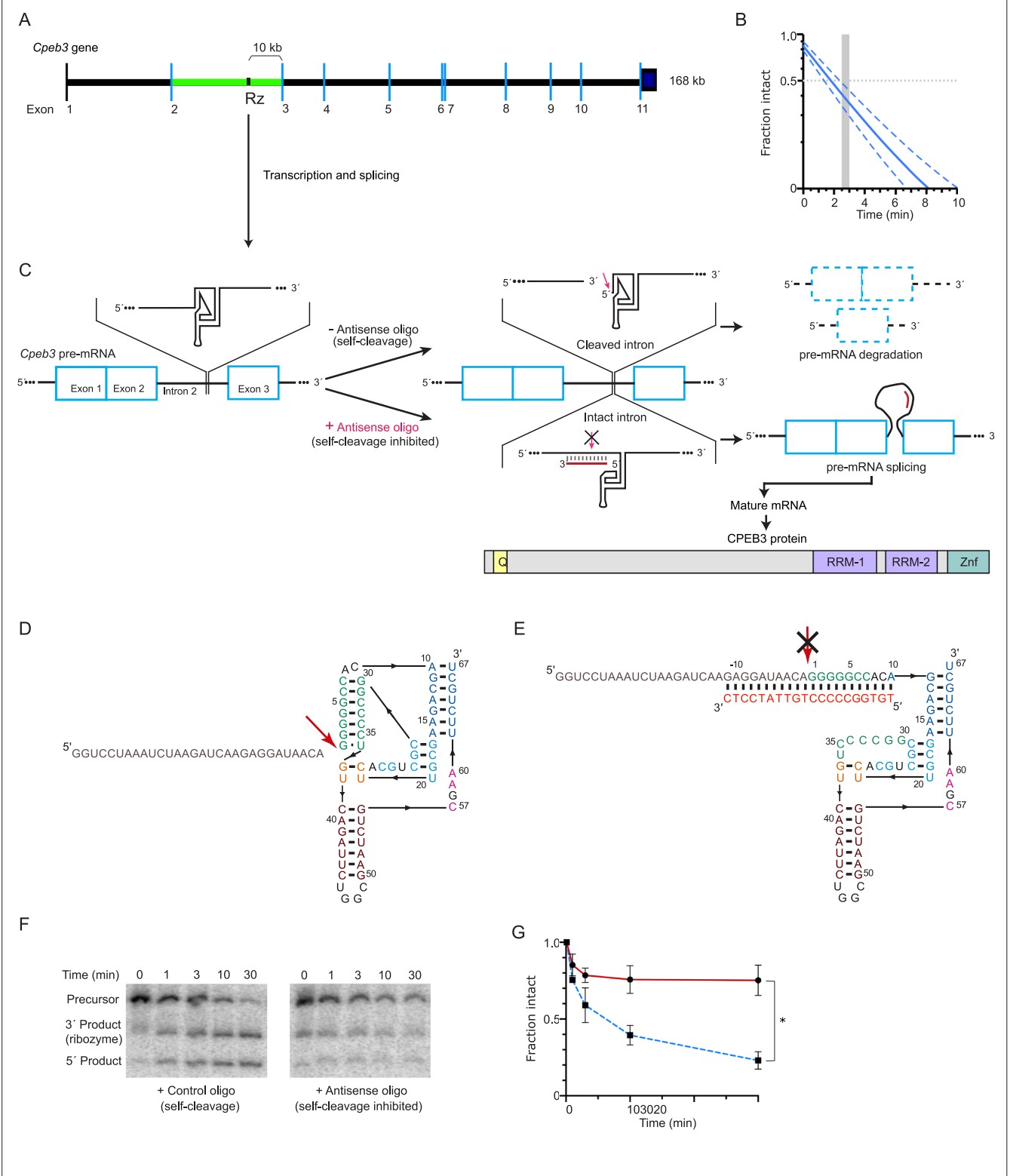

**Figure 1.** *Cpeb3* gene structure and activity of its intronic self-cleaving ribozyme. (**A**) Schematic representation of mouse *Cpeb3* gene. Rz denotes the location of the self-cleaving ribozyme in the second intron (green) between the second and third exons. (**B**) Co-transcriptional self-cleavage activity of a 470-nt construct, incorporating the 72-nt ribozyme, which cuts the transcript 233 nts from the 5′ terminus (see *Table 1* for kinetic parameters of this and other constructs). Log-linear graph of self-cleavage is shown with a solid blue line (dashed lines show ± standard deviation). Gray dotted line indicates midpoint of self-cleavage (with resulting $t_{1/2}$ of ~2 min). Gray bar indicates the approximate time range for RNAPII to travel from the ribozyme to the

*Figure 1 continued on next page*

*Figure 1 continued*

third exon, at which point ~40% of the intron would remain intact. (C) Inhibition of the *Cpeb3* ribozyme by an antisense oligonucleotide (ASO) targeting its cleavage site and the resulting effect on the levels of the spliced mRNA and the encoded protein. (D) Secondary structure of the ribozyme (colored by structural elements; *Webb and Lupták, 2011*). Sequence upstream of the ribozyme is shown in gray, and the site of self-scission is shown with a red arrow. (E) Model of the ribozyme inhibited by the ASO (red letters) showing base-pairing between the ASO and 10 nts upstream and downstream of the ribozyme cleavage site. Inhibition of self-scission is indicated by crossed arrow (C, E). (F) Inhibition of *Cpeb3* ribozyme self-scission in vitro in the presence of ASO. Scrambled or ASO (1 µM) were added during co-transcriptional self-cleavage reactions. (G) Fraction intact values were calculated and plotted vs. time. Significant inhibition of co-transcriptional self-scission by the ASO (red line, compared with control oligo shown in blue), resulting in increase of intact RNA (F, G), is observed at the 3 min time point relevant to the transcription of the *Cpeb3* gene (A, B) (unpaired *t*-test, $t_{(3.599)}$ = 8.204, p=0.0019, *p<0.05; n = 2: control, n = 4: ASO). Data are presented as mean ± SEM.

The online version of this article includes the following source data for figure 1:

**Source data 1.** Source data for *Figure 1B, F, and G*.

**Source data 2.** Full raw unedited PAGE images.

ASOs are synthetic single-stranded nucleic acids that can bind to pre-mRNA or mature RNA by base-pairing, and typically trigger RNA degradation by RNase H. ASOs have also been employed to modulate alternative splicing, suggesting that they act co-transcriptionally in vivo (e.g., to correct the *SMN2* gene; *Hua et al., 2010*). We designed and screened a series of ASOs with the goal of blocking co-transcriptional self-scission of the *Cpeb3* ribozyme. The greatest inhibition was observed when the ASO was bound to the ribozyme cleavage site (*Figure 1C–E*); similar ASOs have been used to inhibit in vitro co-transcriptional self-scission of other HDV-like ribozymes (*Harris et al., 2004*; *Webb et al., 2009*). As the *Cpeb3* ribozyme was synthesized, 80% of it remained uncleaved in the presence of this ASO compared to 20% in the presence of a control oligonucleotide at the 30 min time point (unpaired *t*-test, $t_{(3.599)}$ = 8.204, p=0.0019; *Figure 1F and G*). This ASO and a scrambled control sequence were used in all subsequent in cellulo and in vivo experiments.

## *Cpeb3* mRNA expression is elevated in response to neuronal stimulation

Neuronal activity-dependent gene regulation is essential for synaptic plasticity (*Neves et al., 2008*). To investigate the effect of the *Cpeb3* ribozyme on *Cpeb3* mRNA expression and measure its effect on maturation and protein levels, we began by stimulating primary cortical neurons with glutamate or potassium chloride (KCl). *Cpeb3* mRNA levels were measured using primers that specifically amplified exon–exon splice junctions (exons 2–3, 3–6, and 6–9; *Figure 1A*). We found that membrane depolarization by KCl led to an upregulation of *Cpeb3* mRNA 1–2 hr post-stimulation compared with non-stimulated cultures (exons 2–3: $F_{(5,12)}$ = 18.02, p<0.0001; exons 3–6: $F_{(5,12)}$ = 25.48, p<0.0001; exons 6–9: $F_{(5,12)}$ = 4.376, p=0.0168; one-way ANOVA with Šidák's *post hoc* tests; *Figure 2A*). To examine *Cpeb3* ribozyme activity, total ribozyme and uncleaved ribozyme levels were measured by qRT-PCR using primers designed to amplify the ribozyme sequence downstream of the cleavage site and across the cleavage site, respectively. We used a standard curve specific for every amplicon to independently determine the levels of every RNA segment (determined by each primer pair) measured by qRT-PCR. Our results showed that ribozyme expression is elevated at 1 hr following KCl treatment ($F_{(5,17)}$ =

**Table 1.** Kinetic parameters of murine *Cpeb3* ribozyme constructs[†].

| Construct* | A | $k_1$ | B | $k_2$ | C |
|---|---|---|---|---|---|
| −10/72 | 0.72 ± 0.09 | 0.39 ± 0.09 | | | 0.082 ± 0.026 |
| −49/72/165 | 0.88 ± 0.02 | 0.42 ± 0.04 | 0.013 ± 0.015 | 0.11 ± 0.03 | 0.04 ± 0.02 |
| −233/72/165 | 0.78 ± 0.04 | 0.31 ± 0.04 | 0.035 ± 0.006 | 0.17 ± 0.02 | 0.029 ± 0.005 |

*Construct size is defined as (length of sequence upstream of the ribozyme cleavage site)/[*Cpeb3* ribozyme (72 nts)]/(downstream sequence).

[†]Co-transcriptional self-scission was modeled by a bi-exponential decay model with a residual. A and B represent fractions of the population cleaving with fast ($k_1$) and slow ($k_2$) rate constants, cleave. Errors represent SEM of at least three experiments. For the smallest ribozyme construct (-10/72), a monoexponential decay function was sufficient to model the data.

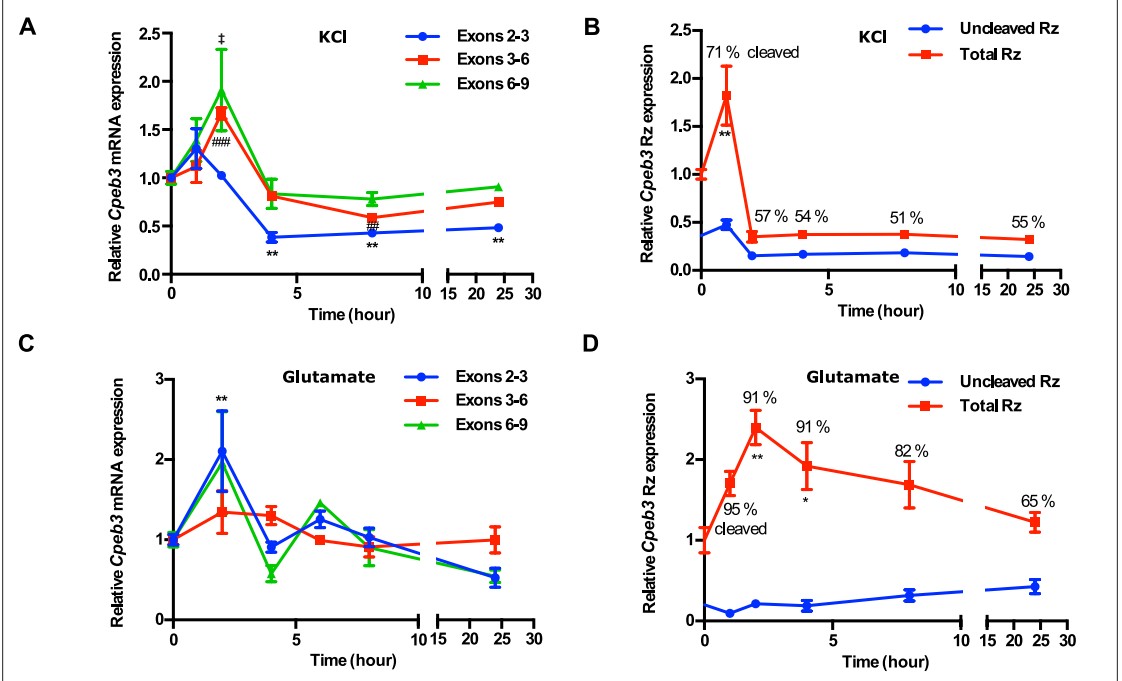

**Figure 2.** *Cpeb3* expression in primary cortical neurons (DIV14). (**A**) KCl stimulation profile of the *Cpeb3* gene showing induction of spliced *Cpeb3* exons (one-way ANOVA, exons 2–3: $F_{(5,12)}$ = 18.02, p<0.0001, Šidák's *post hoc* tests,*p<0.05, **p<0.01; exons 3–6: $F_{(5,12)}$ = 25.48, p<0.0001, Šidák's *post hoc* tests, ##p<0.01, ###p<0.001; exons 6–9: $F_{(5,12)}$ = 4.376, p=0.0168, Šidák's *post hoc* tests, ‡p<0.05. n = 3). (**B**) KCl stimulation profile of *Cpeb3* ribozyme expression (uncleaved and total). Cleaved ribozyme fraction is calculated as [(total ribozyme – uncleaved ribozyme)/total ribozyme] and shown as % cleaved (one-way ANOVA, $F_{(5,17)}$ = 12.96, p<0.0001, Šidák's *post hoc* tests, **p<0.01. n = 6, 3, 3, 3, 3). (**C**) Expression of *Cpeb3* mRNA exons 2–3 is upregulated 2 hr after glutamate stimulation (one-way ANOVA: exons 2–3: $F_{(5,21)}$ = 5.826, p=0.0016, Šidák's *post hoc* tests, **p<0.01; exons 3–6: $F_{(5,22)}$ = 2.002, p=0.1181; exons 6–9: $F_{(5,22)}$ = 1.763, p=0.1622. n = 6, 4, 4, 4, 6, 3). (**D**) Glutamate stimulation induces an increase in *Cpeb3* ribozyme levels at 2 hr time point (one-way ANOVA, $F_{(5,26)}$ = 4.657, p=0.0036, Šidák's *post hoc* test. *p<0.05, **p<0.01. n = 9, 4, 4, 6, 6, 3). Data are presented as mean ± SEM.

The online version of this article includes the following source data and figure supplement(s) for figure 2:

**Source data 1.** Tabulated data for *Figure 2*.

**Figure supplement 1.** Transcriptome analysis of *Cpeb3* gene in the mouse hippocampus.

12.96, p<0.0001; one-way ANOVA with Šidák's *post hoc* tests; *Figure 2B*). Similarly, glutamate stimulation resulted in increased expression of spliced exons by two- to threefold at 2 hr, with a decrease observed at later time points (exons 2–3: $F_{(5,21)}$ = 5.826, p=0.0016; exons 3–6: $F_{(5,22)}$ = 2.002, p=0.1181; exons 6–9: $F_{(5,22)}$ = 1.763, p=0.1622; one-way ANOVA with Šidák's *post hoc* tests; *Figure 2C*), and increased ribozyme expression correlated with *Cpeb3* mRNA expression ($F_{(5,26)}$ = 4.657, p=0.0036; one-way ANOVA with Šidák's *post hoc* tests; *Figure 2D*). This finding is supported by previous studies showing that synaptic stimulation by glutamate leads to an increase in CPEB3 protein expression in hippocampal neurons (*Fioriti et al., 2015*) and that treatment with kainate likewise induces *Cpeb3* expression in the hippocampus (*Theis et al., 2003*). The cleaved fraction of the ribozyme, determined as the difference between the uncleaved fraction and unity, was greatest at the highest point of *Cpeb3* mRNA expression, indicating efficient co-transcriptional self-scission. Furthermore, nuRNA-sequencing analysis of the GSE125068 dataset revealed the *Cpeb3* induction in the mouse hippocampus following kainic acid (KA) administration (*Fernandez-Albert et al., 2019*). The early segments of *Cpeb3* (spanning approximately exons 1–4) exhibited increased expression 1 hr after KA injection compared to the saline group, and the expression levels returned to baseline at 6 and 48 hr post-injection (*Figure 2—figure supplement 1A*). KA, a glutamate receptor agonist, induces neuronal activation in vivo through membrane depolarization and calcium influx. Importantly, analysis of the intron expression around the ribozyme showed that the number of sequencing reads upstream and downstream of the ribozyme cleavage site is elevated at 1 hr post-induction, but no reads spanning the ribozyme cleavage site are observed, supporting the model that the ribozyme self-cleaves co-transcriptionally (*Figure 2—figure supplement 1B*). These data, together with our observations,

suggested that *Cpeb3* expression is activity-dependent, and the *Cpeb3* ribozyme self-cleaves in vivo and potentially *cis*-regulates the maturation of *Cpeb3* mRNA.

## *Cpeb3* mRNA levels increase in primary neuronal cultures treated with ribozyme inhibitor

Because our data showed that *Cpeb3* ribozyme expression and self-scission is correlated with mRNA expression, we hypothesized that modulation of the ribozyme activity may alter *Cpeb3* mRNA splicing. If so, then abrogation of the ribozyme self-scission would result in uncleaved second intron and higher levels of spliced mRNA. We inhibited the ribozyme using ASOs that were designed to increase thermal stability of complementary hybridization and, as a result, induce higher binding affinity for the ribozyme. To study the effect of the *Cpeb3* ribozyme on *Cpeb3* mRNA expression, neuronal cultures were pretreated with either an ASO or a non-targeting (scrambled) control oligonucleotide, followed by KCl stimulation. In the absence of ASO, KCl induced a rapid and robust increase in ribozyme levels compared to cultures containing scrambled ASO. This effect was suppressed in the presence of ASO, which is consistent with the ASO blocking the ribozyme (two-way ANOVA with Šidák's *post hoc* tests, significant main effect of KCl: $F_{(1,19)}$ = 8.058, p=0.0105; significant effect of ASO: $F_{(1,19)}$ = 12.88, p=0.0020; no significant interaction: $F_{(1,19)}$ = 3.557, p=0.0747; *Figure 3A*). At an early time point (2 hr post-KCl induction), the ASO-containing culture displayed an increase of spliced mRNA (exons 2–3: two-way ANOVA with Šidák's *post hoc* tests, significant effect of ASO: $F_{(1,20)}$ = 21.81, p=0.0001, no significant effect of KCl: $F_{(1,20)}$ = 0.1759, p=0.6794; no significant interaction: $F_{(1,20)}$ = 0.001352, p=0.9710; *Figure 3B*; exons 3–6: two-way ANOVA with Šidák's *post hoc* tests, significant ASO × KCl interaction: $F_{(1,19)}$ = 5.726, p=0.0272; significant effect of ASO: $F_{(1,19)}$ = 8.042, p=0.0106; no significant effect of KCl: $F_{(1,19)}$ = 0.2922, p=0.5951; *Figure 3C*; exons 6–9: two-way ANOVA with Šidák's *post hoc* tests, no significant effect of KCl: $F_{(1,19)}$ = 1.218, p=0.2835, no significant effect of ASO: $F_{(1,19)}$ = 3.919, p=0.0624, and no significant interaction: $F_{(1,19)}$ = 0.002317, p=0.9621; *Figure 3D*). The ASO likely prevents *Cpeb3* ribozyme from cleaving the intron co-transcriptionally and thereby promotes mRNA maturation, leading to more spliced mRNA and rapid degradation of the ribozyme-harboring intron. At 24 hr post-KCl induction, we observed no significant difference in *Cpeb3* ribozyme expression among groups (two-way ANOVA with Šidák's *post hoc* tests, no significant effect of KCl: $F_{(1,18)}$ = 0.7897, p=0.3859, no significant effect of ASO: $F_{(1,18)}$ = 0.03687, p=0.8499, and no significant interaction: $F_{(1,18)}$ = 0.9533, p=0.3418; *Figure 3E*). Likewise, the level of *Cpeb3* mRNA exons 2–3 returned to the basal level (two-way ANOVA with Šidák's *post hoc* tests, no significant effect of KCl: $F_{(1,19)}$ = 0.0004856, p=0.9826; no significant effect of ASO: $F_{(1,19)}$ = 3.188, p=0.0902, and no significant interaction: $F_{(1,19)}$ = 0.4343, p=0.5178; *Figure 3F*), while exons 3–6 remained slightly elevated in the ASO-treatment groups (two-way ANOVA with Šidák's *post hoc* tests, significant effect of ASO: $F_{(1,19)}$ = 11.48, p=0.0031; no significant effect of KCl: $F_{(1,19)}$ = 2.252, p=0.1499; no significant interaction: $F_{(1,19)}$ = 0.04047, p=0.8417; *Figure 3G*). The mRNA expression of *Cpeb3* exons 6–9 remained stable over time and was not affected by ASO treatment or KCl stimulation (two-way ANOVA with Šidák's *post hoc* tests, no significant effect of KCl: $F_{(1,19)}$ = 0.6316, p=0.4366; no significant effect of ASO: $F_{(1,19)}$ = 1.364, p=0.2573, and no significant interaction: $F_{(1,19)}$ = 0.1475, p=0.7052; *Figure 3H*).

We further evaluated whether inhibition of *Cpeb3* ribozyme affects the levels of full-length *Cpeb3* mRNA and found that ASO treatment led to a significant increase of spliced exons 2–9 (which correspond to the protein-coding segment of the mRNA) at the 2 hr time point (unpaired *t*-test, $t_{(10.00)}$ = 3.774, p=0.0036; *Figure 3I*). Taken together, these data show that the *Cpeb3* ribozyme modulates the production of the full-length *Cpeb3* mRNA.

To determine whether the ASO specifically targets *Cpeb3* ribozyme or modulates intron levels in general, we measured the levels of the fourth *Cpeb3* intron, which does not harbor a self-cleaving ribozyme. No significant difference in the fourth intron expression was observed between groups, demonstrating that the ASO does not have a broad nonspecific effect on the stability of other introns (two-way ANOVA with Šidák's *post hoc* tests, no significant effect of KCl: $F_{(1,18)}$ = 4.187, p=0.0566; no significant effect of ASO: $F_{(1,18)}$ = 1.032, p=0.3232; no significant interaction: $F_{(1,18)}$ = 0.00001455, p=0.9970; *Figure 3J*). Similarly, we measured mRNA expression of other members of the *Cpeb* gene family (*Cpeb1*, *Cpeb2*, and *Cpeb4*), and our results revealed no significant difference in the gene expression between Ctrl-ASO and ASO groups (*Cpeb1*: $t_{(8.777)}$ = 0.6338, p=0.5423; *Cpeb2*: $t_{(7.768)}$ = 1.491, p=0.1753; *Cpeb4*: $t_{(8.270)}$ = 0.6268, p=0.5477; unpaired *t*-test; *Figure 3K*). These results confirm

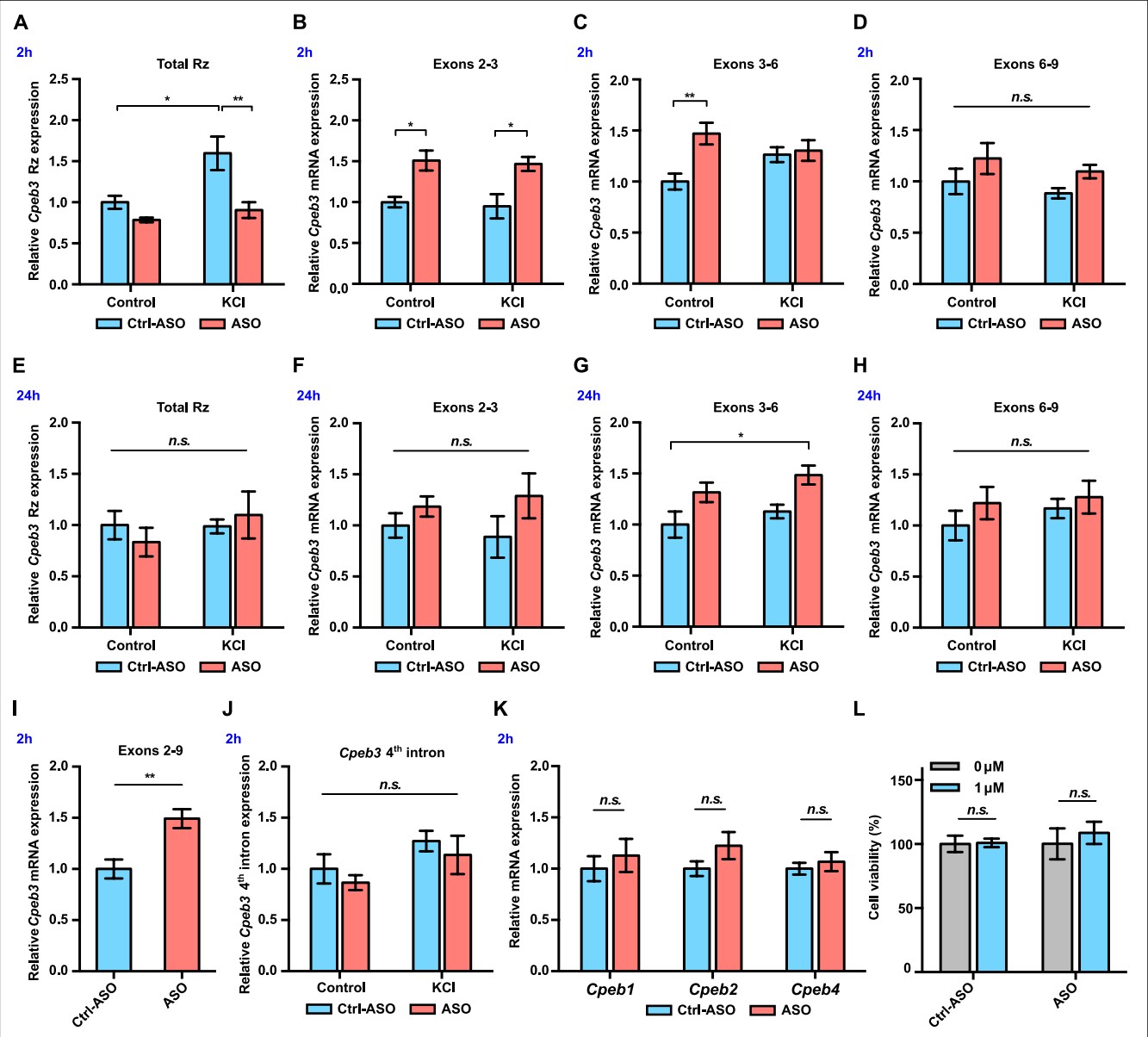

**Figure 3.** *Cpeb3* mRNA is upregulated in primary neuronal cultures (DIV14) treated with ribozyme antisense oligonucleotide (ASO). (**A**) *Cpeb3* ribozyme levels increase together with levels of the surrounding exons 2 hr post-stimulation in experiments with control ASO. Ribozyme levels are significantly lower in ribozyme ASO experiments, suggesting that the RT-PCR reaction is blocked by the ASO (two-way ANOVA with Šidák's *post hoc* tests, significant main effect of KCl: $F_{(1,19)}$ = 8.058, p=0.0105; significant effect of ASO: $F_{(1,19)}$ = 12.88, p=0.0020; no significant interaction: $F_{(1,19)}$ = 3.557, p=0.0747. n = 6). (**B**) Ribozyme inhibition by ASO resulted in upregulation of *Cpeb3* (exons 2–3) mRNA (two-way ANOVA with Šidák's *post hoc* tests, significant ASO × KCl interaction: $F_{(1,19)}$ = 5.726, p=0.0272; significant effect of ASO: $F_{(1,19)}$ = 8.042, p=0.0106; no significant effect of KCl: $F_{(1,19)}$ = 0.2922, p=0.5951. n = 6). (**C**) Inhibition of *Cpeb3* ribozyme by ASO resulted in upregulation of *Cpeb3* mRNA basal levels for exons 3–6 at the 2 hr time point (two-way ANOVA with Šidák's *post hoc* tests, significant ASO × KCl interaction: $F_{(1,19)}$ = 5.726, p=0.0272; significant effect of ASO: $F_{(1,19)}$ = 8.042, p=0.0106; no significant effect of KCl: $F_{(1,19)}$ = 0.2922, p=0.5951 n = 6). (**D**) Levels of exons 6–9 did not increase significantly at the 2 hr time point (two-way ANOVA with Šidák's *post hoc* tests, no significant effect of KCl: $F_{(1,19)}$ = 1.218, p=0.2835, no significant effect of ASO: $F_{(1,19)}$ = 3.919, p=0.0624, and no significant interaction: $F_{(1,19)}$ = 0.002317, p=0.9621). (**E**) No statistically significant difference in *Cpeb3* ribozyme expression was observed after 24 hr post KCl induction, suggesting that all intronic RNA levels reached basal levels (two-way ANOVA with Šidák's *post hoc* tests, no significant effect of KCl: $F_{(1,18)}$ = 0.7897, p=0.3859, no significant effect of ASO: $F_{(1,18)}$ = 0.03687, p=0.8499, and no significant interaction: $F_{(1,18)}$ = 0.9533, p=0.3418. n = 6). (**F–H**) *Cpeb3* mRNA expression largely returned to the basal level 24 hr post-stimulation, although levels of spliced exons 3–6 remained elevated. (**F**) Exons 2–3, two-way ANOVA with Šidák's *post hoc* tests, no significant effect of KCl: $F_{(1,19)}$ = 0.0004856, p=0.9826; no significant effect of ASO: $F_{(1,19)}$ = 3.188, p=0.0902, and no significant interaction: $F_{(1,19)}$ = 0.4343, p=0.5178; n = 6. (**G**) Exons 3–6, two-way ANOVA with Šidák's *post hoc* tests, significant effect of ASO: $F_{(1,19)}$ = 11.48, p=0.0031; no significant effect of KCl: $F_{(1,19)}$ = 2.252, p=0.1499; no significant interaction: $F_{(1,19)}$ = 0.04047, p=0.8417. n = 6. (**H**) Exons 6–9, two-way

*Figure 3 continued on next page*

*Figure 3 continued*

ANOVA with Šidák's *post hoc* tests, no significant effect of KCl: $F_{(1,19)}$ = 0.6316, p=0.4366; no significant effect of ASO: $F_{(1,19)}$ = 1.364, p=0.2573, and no significant interaction: $F_{(1,19)}$ = 0.1475, p=0.7052. n = 6. (**I**) ASO treatment leads to an increase of *Cpeb3* full-length mRNA (exons 2–9, unpaired *t*-test, $t_{(10.00)}$=3.774, p=0.0036. n = 6). (**J**) qRT-PCR analysis of *Cpeb3* fourth intron expression reveals that the ribozyme ASO does not affect its levels, suggesting that it is specific for the ribozyme (two-way ANOVA with Šidák's *post hoc* tests, no significant effect of KCl: $F_{(1,18)}$ = 4.187, p=0.0566; no significant effect of ASO: $F_{(1,18)}$ = 1.032, p=0.3232; no significant interaction: $F_{(1,18)}$ = 0.00001455, p=0.9970. n = 6). (**K**) *Cpeb3* ribozyme ASO does not alter *Cpeb1*, *Cpeb2*, and *Cpeb4* mRNA expression, demonstrating the specificity of the ASO (*Cpeb1*: $t_{(8,777)}$ = 0.6338, p=0.5423; *Cpeb2*: $t_{(7,768)}$ = 1.491, p=0.1753; *Cpeb4*: $t_{(8,270)}$ = 0.6268, p=0.5477; unpaired *t*-test. n = 6). (**L**) Effect of ASO treatment on cell viability. XTT assay was performed after 18 hr incubation of ASOs. Relative cell viability was normalized to the vehicle control ($t_{(2.986)}$ = 0.1257, p=0.9079; ASO: $t_{(5.437)}$ = 0.5869, p=0.5808; unpaired *t*-test. n = 4). *p<0.05, **p<0.01, *n.s.* not significant. Data are presented as mean ± SEM.

The online version of this article includes the following source data for figure 3:

**Source data 1.** Tabulated data for *Figure 3*.

that the ASO is specific for the *Cpeb3* ribozyme and only modulates levels of the *Cpeb3* mRNA. To assess whether the ASO induces cytotoxicity in vitro, neuronal cultures were treated with either ASO or Ctrl-ASO. Cell viability was measured with an XTT assay and revealed no difference in either ASO- or scrambled-ASO-treated cells, compared to untreated cells. Thus, the ASOs used in this study did not induce cytotoxic effects in cultured neurons (Ctrl-ASO: $t_{(2.986)}$ = 0.1257, p=0.9079; ASO: $t_{(5.437)}$ = 0.5869, p=0.5808; unpaired *t*-test; *Figure 3L*).

## Ribozyme inhibition leads to increased expression of CPEB3 and plasticity-related proteins

We next determined whether inhibition of *Cpeb3* ribozyme affects CPEB3 protein expression. Treatment with the ribozyme ASO resulted in a significant increase in CPEB3 protein levels both in the basal state and under KCl-stimulated conditions, indicating a coordination of activity-dependent transcription and translation upon inhibition of *Cpeb3* ribozyme (two-way ANOVA with Šidák's *post hoc* tests, significant effect of ASO: $F_{(1,24)}$ = 21.68, p<0.0001; no significant effect of KCl: $F_{(1,24)}$ = 0.6204, p=0.4386; no significant interaction: $F_{(1,24)}$ = 1.556, p=0.2243; *Figure 4A and B*).

Previous studies have demonstrated the role of CPEB3 in the translational regulation of a number of plasticity-related proteins (PRPs), including AMPA-type glutamate receptors (AMPARs), NMDA receptor (NMDAR), and postsynaptic density protein 95 (PSD-95, product of *Dlg4* gene) (*Huang et al., 2006*; *Chao et al., 2012*; *Chao et al., 2013*; *Fioriti et al., 2015*). As an RNA-binding protein, CPEB3 binds to 3' UTR of *Gria1*, *Gria2*, and *Dlg4* mRNAs and regulates their polyadenylation and translation (*Huang et al., 2006*; *Pavlopoulos et al., 2011*; *Chao et al., 2013*; *Fioriti et al., 2015*). Treatment with the *Cpeb3* ribozyme ASO resulted in a significant increase in GluA1 and PSD-95 protein expression, whereas GluA2 levels remained unchanged (GluA1: two-way ANOVA with Šidák's *post hoc* tests, significant effect of ASO: $F_{(1,24)}$ = 7.134, p=0.134; no significant effect of KCl: $F_{(1,24)}$ = 0.07449, p=0.7872; and no significant interaction: $F_{(1,24)}$ = 1.911, p=0.1796; *Figure 4C and D*; GluA2: two-way ANOVA with Šidák's *post hoc* tests, no significant effect of ASO: $F_{(1,24)}$ = 2.149, p=0.1556; no significant effect of KCl: $F_{(1,24)}$ = 0.04578, p=0.8324; and no significant interaction: $F_{(1,24)}$ = 0.006228, p=0.9358; *Figure 4C and E*; PSD-95: two-way ANOVA with Šidák's *post hoc* tests, significant effect of ASO: $F_{(1,24)}$ = 8.213, p=0.0085; no significant effect of KCl: $F_{(1,24)}$ = 0.4082, p=0.5290; and no significant interaction: $F_{(1,24)}$ = 0.5106, p=0.4818; *Figure 4C and F*). Likewise, ASO treatment led to an upregulation of NR2B protein, which is one of the NMDAR subunits (two-way ANOVA with Šidák's *post hoc* tests, significant effect of ASO: $F_{(1,19)}$ = 10.40, p=0.0045; no significant effect of KCl: $F_{(1,19)}$ = 1.791, p=0.2078; and no significant interaction: $F_{(1,19)}$ = 1.444, p=0.2982; *Figure 4G and H*). Thus, our results demonstrate that *Cpeb3* ribozyme activity affects several downstream processes, particularly mRNA maturation and translation, but also the expression of PRPs, including the translation of AMPAR and NMDAR mRNAs.

## *Cpeb3* ribozyme ASO leads to an increase of *Cpeb3* mRNA and polyadenylation of PRPs in the CA1 hippocampus

To investigate whether the *Cpeb3* ribozyme exhibits similar effects in regulating mRNAs related to synaptic plasticity in vivo, mice were stereotaxically infused with either ribozyme ASO, Ctrl-ASO, or vehicle into the CA1 region of the dorsal hippocampus, a major brain region involved in memory

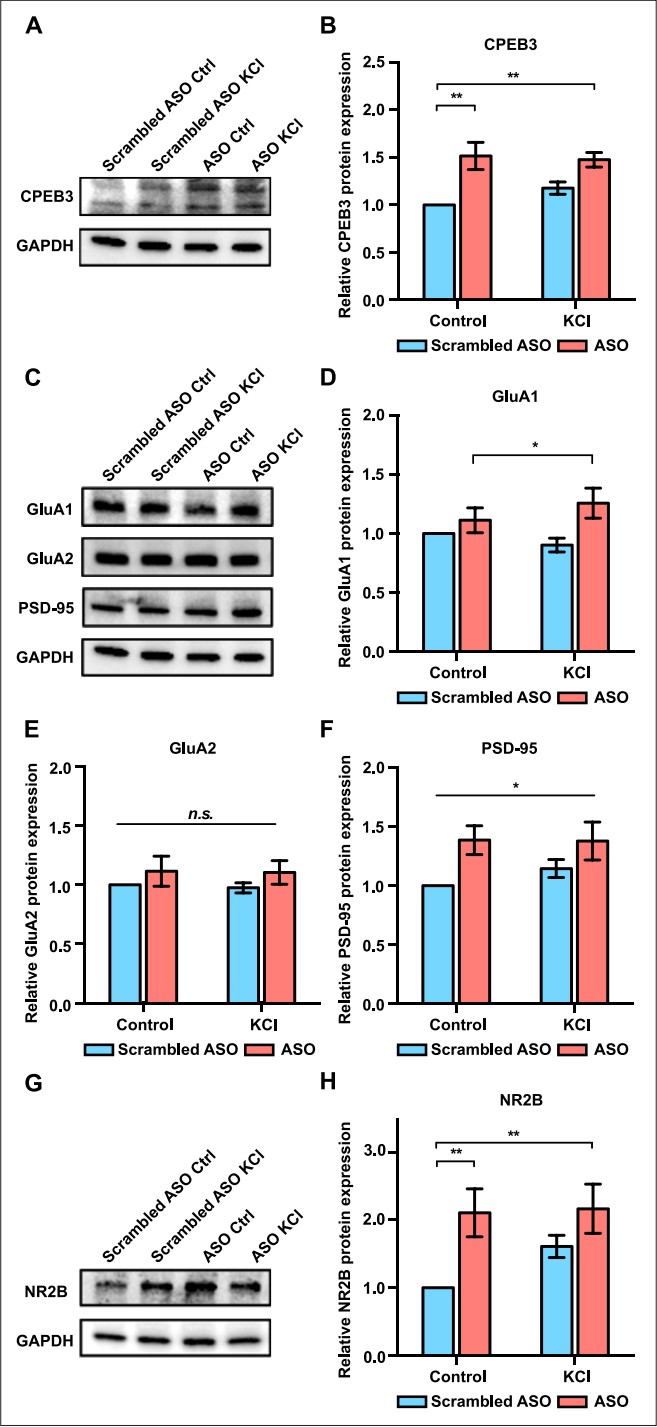

**Figure 4.** Effect of *Cpeb3* ribozyme antisense oligonucleotide (ASO) on protein expression in cultured cortical neurons (DIV7). (**A**) Effect of *Cpeb3* ribozyme ASO on CPEB3 protein expression. Representative image of CPEB3 protein expression. GAPDH is used as a loading control. (**B**) Quantification of CPEB3 protein expression. Treatment of ASO followed by KCl stimulation led to an increase of CPEB3 (two-way ANOVA with Šidák's *post hoc* tests, significant effect of ASO: $F_{(1,24)} = 21.68$, p<0.0001; no significant effect of KCl: $F_{(1,24)} = 0.6204$, p=0.4386; no significant interaction: $F_{(1,24)} = 1.556$, p=0.2243. n = 7). (**C**) Representative immunoblotting image of GluA1, GluA2, and PSD-95 protein expression. GAPDH is used as a loading control. (**D**) Quantification of GluA1 protein expression. GluA1 is upregulated in the presence of ASO combined with neuronal stimulation (two-way ANOVA with Šidák's *post hoc* tests, significant effect of ASO: $F_{(1,24)} = 7.134$, p=0.134; no significant effect of KCl: $F_{(1,24)} = 0.07449$, p=0.7872; and no significant interaction: $F_{(1,24)} = 1.911$, p=0.1796. n = 7). (**E**) Quantification of GluA2

*Figure 4 continued on next page*

*Figure 4 continued*

protein expression. No significant difference was observed between ASO and KCl groups (two-way ANOVA with Šidák's *post hoc* tests, no significant effect of ASO: $F_{(1,24)}$ = 2.149, p=0.1556; no significant effect of KCl: $F_{(1,24)}$ = 0.04578, p=0.8324; and no significant interaction: $F_{(1,24)}$ = 0.006228, p=0.9358. n = 7) (**F**) Treatment with ASO leads to an increase of PSD-95 protein level in primary cortical neurons (two-way ANOVA with Šidák's *post hoc* tests, significant effect of ASO: $F_{(1,24)}$ = 8.213, p=0.0085; no significant effect of KCl: $F_{(1,24)}$ = 0.4082, p=0.5290; and no significant interaction: $F_{(1,24)}$ = 0.5106, p=0.4818. n = 7). (**G**) Representative images of immunoblotting analysis showing NR2B protein expression. GAPDH is used as a loading control. (**H**) Quantification of NR2B protein expression. ASO treatment induces an increase in NR2B expression (two-way ANOVA with Šidák's *post hoc* tests, significant effect of ASO: $F_{(1,19)}$ = 10.40, p=0.0045; no significant effect of KCl: $F_{(1,19)}$ = 1.791, p=0.2078; and no significant interaction: $F_{(1,19)}$ = 1.444, p=0.2982. n = 6). *p<0.05, **p<0.01, *n.s.* not significant. Data are presented as mean ± SEM.

The online version of this article includes the following source data for figure 4:

**Source data 1.** Uncropped western blot images and tabulated data for *Figure 4*.

**Source data 2.** Full raw unedited images.

---

consolidation and persistence (*Figure 5A*). Infusion of the ASO targeting the *Cpeb3* ribozyme significantly reduced ribozyme levels detected by RT-qPCR in the dorsal hippocampus (one-way ANOVA with Šidák's *post hoc* tests; $F_{(2,18)}$ = 3.901, p=0.0391; *Figure 5B*). However, administration of ASO led to an increase of *Cpeb3* mRNA in the CA1 hippocampus (one-way ANOVA with Šidák's *post hoc* tests; exons 2–3: $F_{(2,18)}$ = 6.199, p=0.0089; exons 3–6: $F_{(2,18)}$ = 12.44, p=0.0004; exons 6–9: $F_{(2,17)}$ = 11.03, p=0.0008; *Figure 5C*), confirming that the ASO prevents ribozyme self-scission during *Cpeb3* pre-mRNA transcription and thereby increases *Cpeb3* mRNA levels. To further determine the effect of *Cpeb3* ribozyme in regulating mature mRNA processing, the level of *Cpeb3* exons 2–9 was measured. ASO-infused mice exhibited a significant increase in full-length *Cpeb3* mRNA (one-way ANOVA with Šidák's *post hoc* tests; $F_{(2,17)}$ = 4.385, p=0.0291; *Figure 5D*). In line with our in vitro studies, no significant difference in the ribozyme-free fourth intron levels was observed between mouse hippocampus treated with ASO and vehicle (one-way ANOVA with Šidák's *post hoc* tests; $F_{(2,18)}$ = 0.3663, p=0.6984; *Figure 5E*). We also found no significant difference in the levels of other *Cpeb* mRNAs or degree of protein expression between ASO and control groups (one-way ANOVA with Šidák's *post hoc* tests; *Cpeb1* mRNA: $F_{(2,18)}$ = 0.8203, p=0.4570; *Figure 5F*; *Cpeb2* mRNA: $F_{(2,18)}$ = 2.002, p=0.1641; *Figure 5F*; *Cpeb4* mRNA: $F_{(2,18)}$ = 0.3562, p=0.7052; *Figure 5F*; CPEB1 protein: $t_{(8.942)}$ = 0.4469, p=0.6656; *Figure 5G and H*; CPEB4 protein: $t_{(10.24)}$ = 1.089, p=0.3012; *Figure 5G and H*). These findings demonstrate that the ASO used in this study targets the *Cpeb3* ribozyme in vivo with high specificity.

Next, we tested whether the *Cpeb3* ribozyme inhibition affects *Cpeb3* translation. The CPEB3 protein levels in the hippocampus were measured using western blot analysis and revealed elevated CPEB3 protein expression in ASO-treated mice, suggesting that increased translation of *Cpeb3* directly results from increased levels of full-length mRNA ($t_{(14.50)}$ = 2.709, p=0.0165; unpaired *t*-test; *Figure 5I and J*). Furthermore, blocking the *Cpeb3* ribozyme does not change *Gria1*, *Gria2*, *Dlg4*, and *Grin2b* mRNA or protein expression in naïve, home cage mice (GluA1: $t_{(5.848)}$ = 1.655, p=0.1503; GluA2: $t_{(10.96)}$ = 0.5476, p=0.5949; PSD-95: $t_{(8.760)}$ = 0.9838, p=0.3516; NR2B: $t_{(11.11)}$ = 1.250, p=0.2369; *Figure 5K*; GluA1: $t_{(13.18)}$ = 0.6339, p=0.5370; GluA2: $t_{(17.54)}$ = 0.5755, p=0.5723; PSD-95: $t_{(14.94)}$ = 0.8612, p=0.4027; NR2B: $t_{(16.34)}$ = 0.2604, p=0.7978; unpaired *t*-test; *Figure 5L and M*). Thus, in naïve mice, ribozyme inhibition leads to increased basal levels of the *Cpeb3* mRNA and protein, but its downstream mRNA targets remain unchanged in the absence of activity-dependent learning or stimulation.

The *Cpeb3* ribozyme activity may result from polyadenylation of its target mRNAs; therefore, 3′ rapid amplification of cDNA ends (3′ RACE) was performed to examine the 3′ termini of several mRNAs. We found that ribozyme ASO administration led to increased *Gria1*, *Gria2*, and *Dlg4* mRNA polyadenylation in the mouse dorsal hippocampus (*Gria1:* $t_{(10.44)}$ = 2.535, p=0.0287; *Gria2:* $t_{(11.02)}$ = 2.327, p=0.0400; *Dlg4:* $t_{(9.808)}$ = 4.254, p=0.0018; NR2B: $t_{(8.020)}$ = 0.9846, p=0.3536; unpaired *t*-test; *Figure 5N*). These data support a model wherein the inhibition of the *Cpeb3* ribozyme leads to increased polyadenylation of existing AMPARs and *Dlg4* mRNAs, and suggests a role for the ribozyme in post-transcriptional regulation and 3′ mRNA processing.

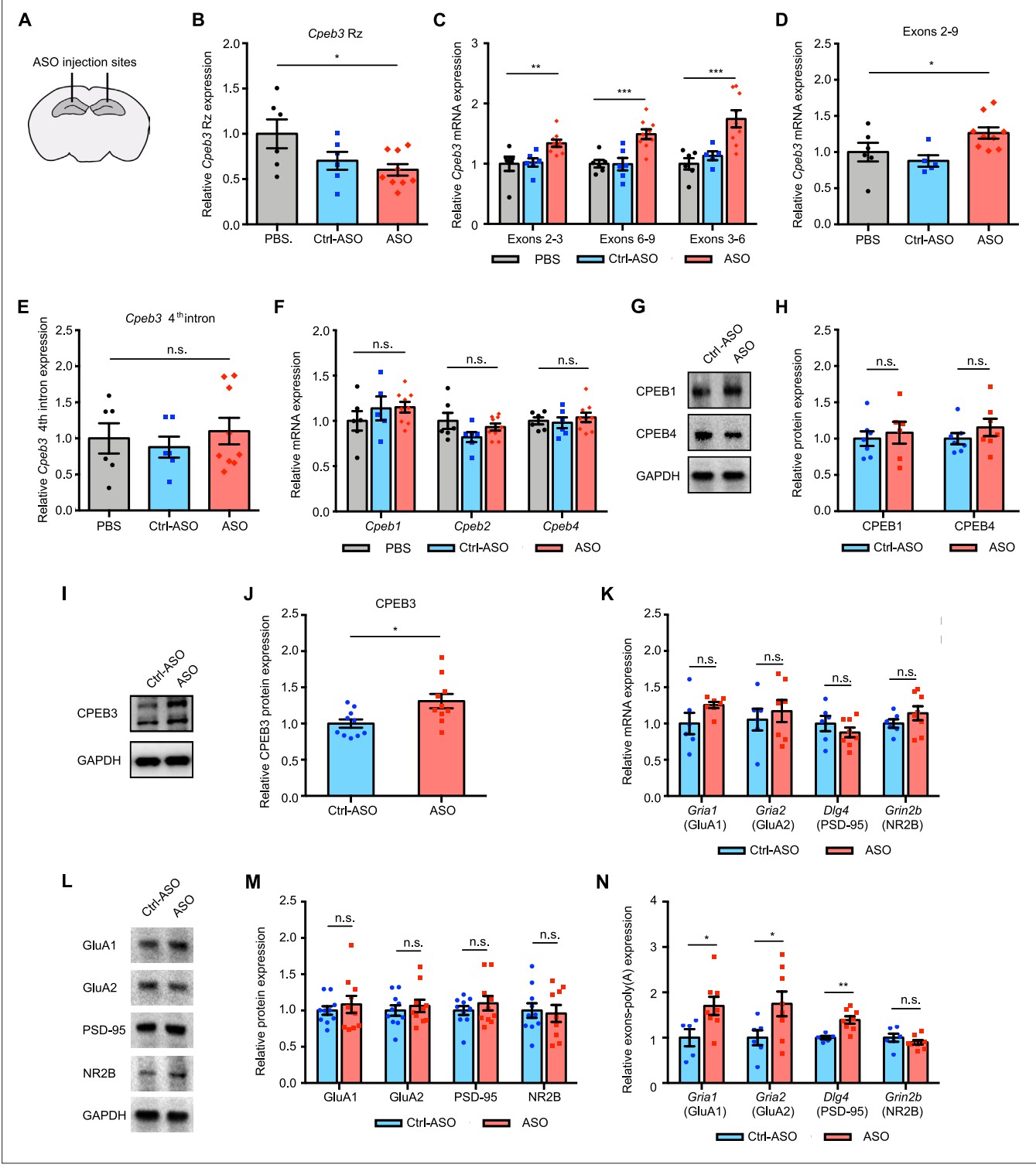

**Figure 5.** *Cpeb3* ribozyme antisense oligonucleotide (ASO) leads to an increase of *Cpeb3* mRNA and polyadenylation of plasticity-related proteins (PRPs) in the CA1 hippocampus. (**A**) Schematic representation of stereotaxic procedure. ASO, Ctrl-ASO, or vehicle was bilaterally infused to the mouse CA1 hippocampus. (**B**) Administration of *Cpeb3* ribozyme ASO to the mouse CA1 hippocampus leads to a decrease in *Cpeb3* ribozyme levels (one-way ANOVA with Šidák's *post hoc* tests; $F_{(2,18)}$ = 3.901, p=0.0391. n = 6 [vehicle], 6 [Ctrl-ASO], 9 [ASO]). (**C**) *Cpeb3* mRNA expression is upregulated in the *Cpeb3* ribozyme ASO treatment group compared to controls (one-way ANOVA with Šidák's *post hoc* tests; exons 2–3: $F_{(2,18)}$ = 6.199, p=0.0089; exons 3–6: $F_{(2,18)}$ = 12.44, p=0.0004; exons 6–9: $F_{(2,17)}$ = 11.03, p=0.0008; n = 6, 6, 9). (**D**) *Cpeb3* full-length mRNA (exons 2–9) is significantly elevated in the presence of ASO (one-way ANOVA with Šidák's *post hoc* tests; $F_{(2,17)}$ = 4.385, p=0.0291 n = 6, 6, 9). (**E**) The *Cpeb3* ribozyme ASO has high specificity

*Figure 5 continued on next page*

*Figure 5 continued*

for its cleavage site (in the third intron) in vivo. qRT-PCR analysis of the fourth intron of *Cpeb3* gene demonstrates no significant difference between controls and ASO groups (one-way ANOVA with Šidák's *post hoc* tests; $F_{(2,18)}$ = 0.3663, p=0.6984. n = 6, 6, 9). (**F**) qRT-PCR analysis reveals no significant difference between controls and ASO groups in *Cpeb1*, *Cpeb2*, and *Cpeb4* mRNA expression (*Cpeb1* mRNA: $F_{(2,18)}$ = 0.8203, p=0.4570; *Cpeb2* mRNA: $F_{(2,18)}$ = 2.002, p=0.1641; *Cpeb4* mRNA: $F_{(2,18)}$ = 0.3562, p=0.7052. n = 6, 6, 9). (**G**) Effect of *Cpeb3* ribozyme on CPEB1 and CPEB4 protein expression. GAPDH is used as a loading control. (**H**) Quantification of CPEB1 and CPEB4 protein expression. *Cpeb3* ribozyme ASO does not change CPEB1 and CPEB4 protein expression (CPEB1 protein: $t_{(8.942)}$ = 0.4469, p=0.6656; CPEB4 protein: $t_{(10.24)}$ = 1.089, p=0.3012. n = 7). (**I**) Effect of *Cpeb3* ribozyme on CPEB3 protein expression. Representative image of immunoblotting analysis. GAPDH is used as a loading control. (**J**) Quantification of CPEB3 protein expression. *Cpeb3* ribozyme ASO leads to an increase of CPEB3 protein expression in the CA1 hippocampus ($t_{(14.50)}$ = 2.709, p=0.0165; unpaired *t*-test. n = 10) (**L, M**). (**K**) Inhibition of *Cpeb3* ribozyme does not affect transcription of other plasticity-related genes. qRT-PCR analysis of mature GluA1, GluA2, PSD-95, and NR2B mRNAs. No significant difference between ASO and control was observed for splice junctions within the mRNAs, showing that modulation of the *Cpeb3* ribozyme does not affect transcription or splicing of these mRNAs (GluA1: $t_{(5.848)}$ = 1.655, p=0.1503; GluA2: $t_{(10.96)}$ = 0.5476, p=0.5949; PSD-95: $t_{(8.760)}$ = 0.9838, p=0.3516; NR2B: $t_{(11.11)}$ = 1.250, p=0.2369. n = 6–7). (**L**) Effect of *Cpeb3* ribozyme on PRP protein expression. Representative images of immunoblotting analysis. GAPDH is used as a loading control. (**M**) Quantification of PRP protein expression. Blocking *Cpeb3* ribozyme does not affect PCPs protein expression in the naïve state (GluA1: $t_{(13.18)}$ = 0.6339, p=0.5370; GluA2: $t_{(17.54)}$ = 0.5755, p=0.5723; PSD-95: $t_{(14.94)}$ = 0.8612, p=0.4027; NR2B: $t_{(16.34)}$ = 0.2604, p=0.7978; unpaired *t*-test. n = 10). (**N**) Inhibition of *Cpeb3* ribozyme resulted in increased polyadenylation of plasticity-related genes (*Gria1*: $t_{(10.44)}$ = 2.535, p=0.0287; *Gria2*: $t_{(11.02)}$ = 2.327, p=0.0400; *Dlg4*: $t_{(9.808)}$ = 4.254, p=0.0018; *Grin2b*: $t_{(8.020)}$ = 0.9846, p=0.3536; unpaired *t*-test. n = 6, 8). *p<0.05, **p<0.01, ***p<0.001, *n.s.* not significant. Data are presented as mean ± SEM.

The online version of this article includes the following source data for figure 5:

**Source data 1.** Uncropped western blot images and tabulated data for *Figure 5*.

**Source data 2.** Full raw unedited images.

## Inhibition of *Cpeb3* ribozyme in the dorsal hippocampus enhances long-term memory

Previous studies have shown that *Cpeb3* is regulated by synaptic activity; for example, MWM training and contextual fear conditioning induced an increase in CPEB3 protein expression, and *Cpeb3* mRNA was upregulated 2 hr after kainate injection (*Theis et al., 2003*). To examine whether *Cpeb3* mRNA is modulated by behavioral training, we subjected mice to an object location memory (OLM) task (*Vogel-Ciernia and Wood, 2014*; *Fioriti et al., 2015*) and isolated hippocampal tissues 1 hr after training (*Figure 6A*). The OLM task has been widely used to study hippocampal-dependent spatial memory. The task is based on an animal's innate preference for novelty and its capability for discriminating spatial relationships between novel and familiar object locations (*Vogel-Ciernia and Wood, 2014*). The OLM and object recognition memory (ORM) tasks were originally introduced in the study of rat memory assessment that relies on the rodents' intrinsic novelty preference rather than conventional reinforcement (*Ennaceur and Delacour, 1988*). We first examined the effect of training on *Cpeb3* mRNA expression. *Cpeb3* mRNA exons 1–2, which span about 33 kb of the gene downstream of the promoter (*Figure 1A*), were upregulated 1 hr after training compared to naïve mice (exons 1–2: $t_{(4.991)}$ = 3.085, p=0.0274; *Figure 6B*). We also observed a slight increase in *Cpeb3* mRNA exons 2–3 in OLM-trained mice compared to naïve mice (exons 2–3: $t_{(7.895)}$ = 1.997, p=0.0814; *Figure 6C*). The two-tailed *t*-test yielded a p-value of 0.0814, whereas the one-tailed *t*-test yielded a p-value of 0.0407. Our primary hypothesis was to assess whether *Cpeb3* exons 2–3 are upregulated by OLM training, as we observed in exons 1–2. While the two-tailed test indicates that the difference is not statistically significant at the conventional alpha level of 0.05, the one-tailed test suggests a marginal significance, with evidence supporting an upregulation of *Cpeb3* mRNA expression by OLM training. Furthermore, to test whether the *Cpeb3* ribozyme is regulated by the behavioral paradigm, we measured the ribozyme expression and self-scission by qRT-PCR and found that OLM training induced *Cpeb3* ribozyme expression ($t_{(6.266)}$ = 3.067, p=0.0208; *Figure 6D*) but no significant difference in ribozyme self-scission between naïve and trained mice was observed ($t_{(6.256)}$ = 1.234, p=0.2616; *Figure 6E*). These results suggest the OLM training modulates *Cpeb3* levels, but the ribozyme activity is not affected by the training.

Although a previous study reported that the *Cpeb3* mRNA level (exons 2–6) was not altered after a MWM test (*Fioriti et al., 2015*), these seemingly contradictory results can be explained by the time points and segments of the mRNA analyzed. The distance from the 5′ terminus of the pre-mRNA and exon 2 is about 33 kb, whereas exon 6 is more than three times farther (110 kb). As a result, RNAP II and the splicing machinery require at least three times longer to produce the spliced exons 2–6 of

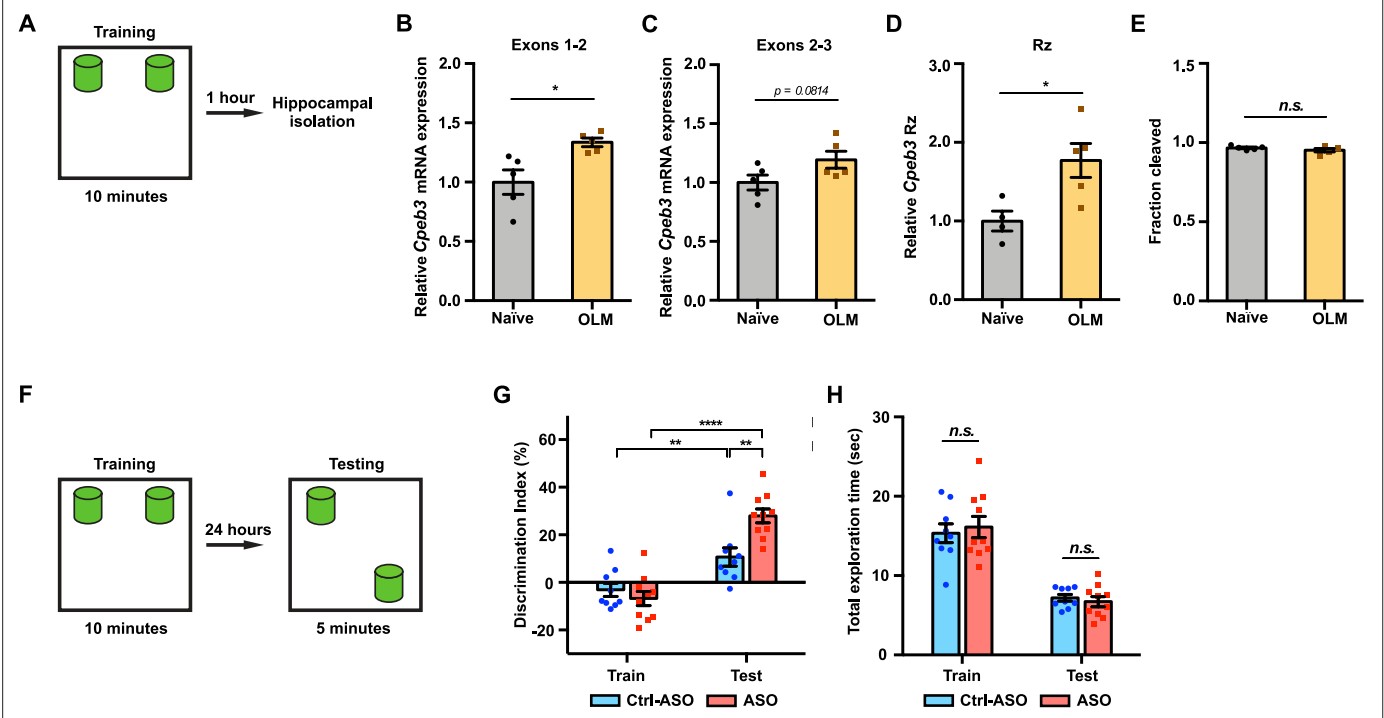

**Figure 6.** Inhibition of *Cpeb3* ribozyme enhances long-term object location memory (OLM). (**A**) Schematic representation of how the hippocampal gene expression is examined after OLM training task. (**B**) OLM training induces expression of *Cpeb3* mRNA exons 1–2 in the CA1 hippocampus ($t_{(4.991)}$ = 3.085, p=0.0274. n = 5). (**C**) OLM training induces a slight upregulation of *Cpeb3* mRNA exons 2–3 in the CA1 hippocampus ($t_{(7.895)}$ = 1.997, p=0.0814. n = 5). (**D**) The *Cpeb3* ribozyme expression is elevated in OLM-trained mice compared to naïve mice ($t_{(6.266)}$ = 3.067, p=0.0208. n = 5). (**E**) The cleaved fraction of the *Cpeb3* ribozyme showed no significant differences between OLM-trained and naïve mice ($t_{(6.256)}$ = 1.234, p=0.2616. n = 5). (**F**) Experimental procedure testing long-term memory. (**G**) Mice infused with Ctrl-ASO or *Cpeb3* ribozyme antisense oligonucleotide (ASO) showed no preference for either object in OLM training. Mice infused with *Cpeb3* ribozyme ASO show significant discrimination index in OLM testing (ASO × session interaction $F_{(1,34)}$ = 11.06, p=0.0021; two-way ANOVA with Šidák's *post hoc* tests. n = 10). (**H**) *Cpeb3* ribozyme ASO and control mice display similar total exploration time (train: $t_{(17.00)}$ = 0.2342, p=0.8176; test: $t_{(13.48)}$ = 1.644, p=0.1232; unpaired *t*-test. n = 10). *p<0.05, **p<0.01, ****p<0.0001, *n.s.* not significant. Data are presented as mean ± SEM.

The online version of this article includes the following source data for figure 6:

**Source data 1.** Tabulated data for *Figure 6*.

the *Cpeb3* mRNA (assuming no significant pausing in transcription and co-transcriptional splicing). Transcription initiation, pre-mRNA production up to exon 2, and splicing would be expected to yield spliced mRNA exons 1–2 after 1 hr, but reaching the sixth exon and splicing the mRNA would likely not happen in that time frame (as evidenced by the GSE125068 nuRNA-seq dataset described above). We therefore believe the results of these two studies are not at odds; rather, these results demonstrate that the detection of new rounds of gene expression should rely on measurements of early segments of activity-induced genes, rather than later segments.

To assess whether inhibition of the *Cpeb3* ribozyme improves memory formation, we studied the effect of the ASO on long-term memory formation for object location using the OLM task (*Figure 6F*). This task requires the dorsal CA1 (*Barrett et al., 2011*; *McQuown et al., 2011*). We therefore infused mice bilaterally into the CA1 dorsal hippocampus with the *Cpeb3* ribozyme ASO, Ctrl-ASO, or vehicle 48 hr prior to OLM training. Mice exhibit no preference for either object, as demonstrated by the absence of significant difference in training discrimination index (DI) ($t_{(16.99)}$ = 0.8967, p=0.3824; unpaired *t*-test; *Figure 6G*). Likewise, during training and testing sessions, similar total exploration times were observed for ASO-infused mice and control mice, demonstrating that both groups of mice have similar exploitative behavior and that the ASO did not simply affect locomotor or exploration performance (train: $t_{(17.00)}$ = 0.2342, p=0.8176; test: $t_{(13.48)}$ = 1.644, p=0.1232; unpaired *t*-test; *Figure 6H*). During the testing session, both Ctrl-ASO- and ASO-treated mice exhibited a significant increase in DI, suggesting that mice exhibited preference in exploring the novel object (Ctrl-ASO: $t_{(14.55)}$

= 2.913, p=0.0110; ASO: $t_{(17.99)}$ = 8.244, p<0.0001; unpaired $t$-test; *Figure 6G*). Notably, the *Cpeb3* ribozyme ASO mice showed a significant increase in DI between training and testing compared to control groups, suggesting that these mice experienced a robust enhancement of novel object exploration (ASO × session interaction $F_{(1,34)}$ = 11.06, p=0.0021; two-way ANOVA with Šidák's *post hoc* tests; *Figure 6G*). An increased preference for exploring the novel object location indicates successful recognition of the spatial change and demonstrates intact spatial memory. In our OLM task protocol, we measured the exploration time when mouse's nose is within 1 cm of the object and directed toward the object, whereas other OLM protocols utilize a 2 cm distance from the object to define the exploration time (*Dix and Aggleton, 1999*), leading to somewhat different exploration times. While different OLM protocols utilize various parameters, and different scoring methods yield different overall exploration times, the calculation of DIs to interpret memory formation from performance remains remarkably stable and the OLM exploration times are similar to previous studies (*Vogel-Ciernia and Wood, 2014*; *Kwapis et al., 2018*; *Shu et al., 2018*; *Kwapis et al., 2019*; *Keiser et al., 2021*; *Dong et al., 2022*). Our results provide strong evidence that *Cpeb3* is critical for long-term memory, and that the *Cpeb3* ribozyme activity is anticorrelated with the formation of long-term memory.

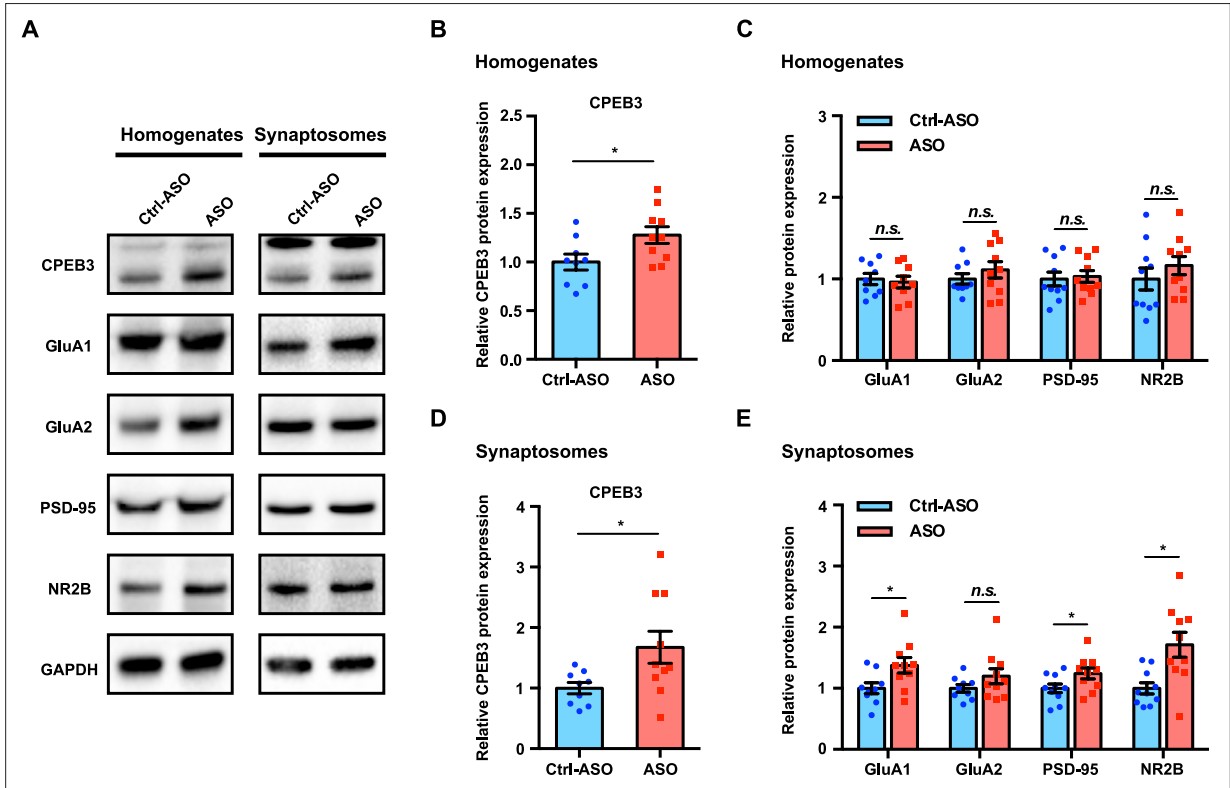

**Figure 7.** Inhibition of *Cpeb3* ribozyme leads to upregulation of CPEB3 and plasticity-related protein (PRP) expression after object location memory (OLM). (**A**) Representative images of immunoblotting analysis. GAPDH is used as a loading control. (**B, C**) Quantification of CPEB3 (**B**) and PRPs (**C**) in tissue homogenates shows increased expression of CPEB3 but not of PRPs (CPEB3: $t_{(17.00)}$ = 2.345, p=0.0314; GluA1: $t_{(15.96)}$ = 0.3751, p=0.7125; GluA2: $t_{(15.16)}$ = 0.9432, p=0.3604; PSD-95: $t_{(17.63)}$ = 0.2849, p=0.7790; NR2B: $t_{(17.32)}$ = 0.9415, p=0.3594; unpaired $t$-test. n = 10). (**D, E**) In synaptosomes, the protein expression of both CPEB3 (**D**) and PRPs (**E**) is increased (CPEB3: $t_{(11.11)}$ = 2.403, p=0.0349; GluA1: $t_{(15.83)}$ = 2.433, p=0.0272; GluA2: $t_{(14.40)}$ = 1.497, p=0.1559; PSD-95: $t_{(17.25)}$ = 2.115, p=0.0493; NR2B: $t_{(12.42)}$ = 3.174, p=0.0077; unpaired $t$-test. n = 10). *p<0.05, *n.s.* not significant. Data are presented as mean ± SEM.

The online version of this article includes the following source data for figure 7:

**Source data 1.** Uncropped western blot images and tabulated data for *Figure 7*.

**Source data 2.** Full raw unedited images.

## *Cpeb3* ribozyme ASO leads to an increase in protein expression of CPEB3 and PRPs during memory consolidation

Learning-induced changes in gene expression and protein synthesis are essential for memory formation and consolidation (*Kandel, 2001*). To determine whether upregulation of *Cpeb3* mRNA by the ribozyme ASO leads to a change in expression of the CPEB3 protein and its downstream targets, we analyzed the dorsal hippocampal homogenates and synaptosomal fractions. Administration of *Cpeb3* ribozyme ASO led to a significant increase of CPEB3 protein expression in the CA1 hippocampal homogenates and crude synaptosomes 1 hr after OLM testing (hippocampal homogenates: $t_{(17.00)}$ = 2.345, p=0.0314; crude synaptosomes: $t_{(11.11)}$ = 2.403, p=0.0349; unpaired *t*-test; *Figure 7A, B, and D*). This result confirms that blocking the *Cpeb3* ribozyme facilitates *Cpeb3* mRNA processing and translation. In addition, the protein levels of GluA1, GluA2, PSD-95, and NR2B were measured to determine whether increased CPEB3 further regulates translation of PRPs. In total tissue lysates, no significant difference in PRP levels was observed between ASO and control (GluA1: $t_{(15.96)}$ = 0.3751, p=0.7125; GluA2: $t_{(15.16)}$ = 0.9432, p=0.3604; PSD-95: $t_{(17.63)}$ = 0.2849, p=0.7790; NR2B: $t_{(17.32)}$ = 0.9415, p=0.3594; unpaired *t*-test; *Figure 7A and C*). However, in synaptosomal fractions, GluA1, PSD-95, and NR2B protein levels were increased in ASO-infused mice, relative to Ctrl-ASO animals; the GluA2 protein level was unaffected (GluA1: $t_{(15.83)}$ = 2.433, p=0.0272; GluA2: $t_{(14.40)}$ = 1.497, p=0.1559; PSD-95: $t_{(17.25)}$ = 2.115, p=0.0493; NR2B: $t_{(12.42)}$ = 3.174, p=0.0077; unpaired *t*-test; *Figure 7A and E*). Our findings thus show that blocking *Cpeb3* ribozyme activity leads to an increase in CPEB3 protein production, and upregulation of CPEB3 by OLM further causes an increase in local GluA1, PSD-95, and NR2B translation.

## Discussion

Self-cleaving ribozymes are broadly distributed small functional RNAs that promote an intramolecular, site-specific, self-scission reaction (*Buzayan et al., 1986*; *Hutchins et al., 1986*; *Prody et al., 1986*; *Sharmeen et al., 1988*; *Saville and Collins, 1990*; *Jimenez et al., 2015*; *Peng et al., 2021*). Despite distinct structures and cut sites, these natural self-cleaving ribozymes all accelerate the same transesterification reaction, which operates via an acid–base catalysis mechanism: nucleophilic attack of a ribose 2'-oxyanion on the adjacent phosphodiester bond yields a 2',3'- cyclic phosphate and a 5'-hydroxyl product (*Wu et al., 1989*; *Fedor, 2009*; *Jimenez et al., 2015*; *Wilson et al., 2016*; *Ren et al., 2017*; *Seith et al., 2018*; *Peng et al., 2021*). Self-cleaving ribozymes act in *cis* (i.e., cut their own backbone) and therefore execute a single catalytic turnover. To date, 10 distinct families of self-cleaving ribozymes have been discovered (*Peng et al., 2021*), but relatively little is known about their biological roles.

The HDV family of ribozymes has been extensively studied: crystal structures have been elucidated, and the mechanism of self-scission (based on a general acid–base catalysis) is well established (*Ferré-D'Amaré et al., 1998*; *Ke et al., 2004*; *Das and Piccirilli, 2005*; *Chen et al., 2010*; *Koo et al., 2015*). These ribozymes operate during rolling circle replication of the HDV RNA genome and in processing of certain non-LTR retrotransposons (*Sharmeen et al., 1988*; *Wu et al., 1989*; *Eickbush and Eickbush, 2010*; *Ruminski et al., 2011*; *Sánchez-Luque et al., 2011*), but given their broad distribution in nature, their biological roles remain largely unexplored. Mammals harbor several self-cleaving ribozymes, all with unknown biological functions (*Salehi-Ashtiani et al., 2006*; *Martick et al., 2008*; *de la Peña and García-Robles, 2010*; *Perreault et al., 2011*; *Hernandez et al., 2020*; *Chen et al., 2021*). One of these ribozymes, the HDV-like *Cpeb3* ribozyme, which is a functionally conserved self-cleaving RNA (*Bendixsen et al., 2021*), maps to the second intron of the *Cpeb3* gene (*Figure 1A*), and its in vitro activity (*Figure 1B*) suggests that its self-scission may be tuned to disrupt the intron at a rate that is similar to the production speed of the downstream intronic sequence ahead of the next exon. Given that the self-scission of intronic ribozymes is inversely correlated with splicing efficiency of the harboring pre-mRNA (*Fong et al., 2009*), we investigated how the endogenous intronic ribozyme affects the *Cpeb3* mRNA maturation and translation, and how it affects memory formation in mice.

Modifications of synaptic strength are thought to underlie learning and memory in the brain. Studies in hippocampal slices revealed local translation in dendrites following induction of LTP (*Frey and Morris, 1997*). Cytoplasmic polyadenylation-induced translation is one of the key steps critical to controlling protein synthesis and neuroplasticity (*Du and Richter, 2005*; *Richter, 2007*; *Richter,*

*2010*), and one of the proteins involved in regulating cytoplasmic polyadenylation of mRNAs is CPEB3. In *Aplysia* sensory-motor neuron co-culture, application of repeated pulses of serotonin (5-HT) induces ApCPEB protein expression at the stimulated synapses and, as a result, LTF, which is a form of learning-related synaptic plasticity that is widely studied in *Aplysia* (*Si et al., 2003*; *Si et al., 2010*). In murine primary hippocampal neurons, the level of CPEB3 protein expression is positively regulated by neuronal activity (*Fioriti et al., 2015*) and plays dual roles in regulating mRNA translation (*Du and Richter, 2005*; *Stephan et al., 2015*): a post-translational modification of CPEB3 (monoubiquitination by Neuralized1) converts it from a repressor to an activator (*Pavlopoulos et al., 2011*).

Polyadenylation-induced translation was first characterized in *Xenopus* oocytes during early development, where untranslated mRNAs possessed short polyA tails; upon exposure to progesterone, the polyA tails were elongated, leading to the initiation of translation (*Richter, 1999*; *Mendez et al., 2000*). In hippocampal neurons, it was suggested that the 3′ untranslated region of mRNA of α-calmodulin-dependent protein kinase II (α-CaMKII) was regulated by CPEB, undergoing polyadenylation-induced translation upon synaptic activation. Further, light exposure-triggered dark-reared rats exhibit significant experience-dependent activity in the visual cortex, where α-CaMKII mRNA was polyadenylated and translated during visual experience (*Wu et al., 1998*). In addition, activation of CPEB3 through Neuralized1 resulted in polyadenylation and translational activity of GluA1 and GluA2 and dendritic formation, which is important for facilitating synaptic transmission (*Pavlopoulos et al., 2011*). These studies underscore the significance of understanding the mechanisms governing polyadenylation-induced translation in synaptic plasticity. Because synaptic local translation is essential for LTM, the modulation of translational process serves a pivotal role for the regulation of synaptic plasticity and memory consolidation.

Several studies have shown that CPEB3 is essential for synaptic strength, regulating mRNA translation of several PRPs at synapses (*Huang et al., 2006*; *Pavlopoulos et al., 2011*; *Fioriti et al., 2015*). Previous reports have shown that CPEB3 regulates GluA1 and GluA2 polyadenylation: *Cpeb3* conditional knockout mice fail to elongate the poly(A) tail of *Gria1* and *Gria2* mRNA after MWM training, and overexpression of CPEB3 changes the length of the *Gria1* and *Gria2* mRNA poly(A) tail (*Fioriti et al., 2015*). Hippocampal-dependent learning and memory is modulated by CPEB3 on the level of translation (*Pavlopoulos et al., 2011*), but it is unknown whether the CPEB3 expression is modulated by the *Cpeb3* ribozyme.

In mammals, the coordination of pre-mRNA processing and transcription can affect gene expression (*Neugebauer, 2019*). Using long-read sequencing and Precision Run-On sequencing (PRO-seq) approaches, measurements of co-transcriptional splicing events in mammalian cells demonstrated that co-transcriptional splicing efficiency impacts productive gene output (*Reimer et al., 2021*). The temporal and spatial window shows that the splicing and transcription machinery are tightly coupled. Our study is agreement with this co-transcriptional splicing model and shows that inhibition of the intronic *Cpeb3* ribozyme leads both to an increase in *Cpeb3* mRNA and protein levels in primary cortical neurons and the dorsal hippocampus upon synaptic stimulation, and subsequently, to changes in the polyadenylation of target mRNAs of the CPEB3 protein.

Activity-dependent synaptic changes are governed by AMPAR trafficking, and AMPARs are mobilized to the postsynaptic surface membrane in response to neuronal activity in a dynamic process (*Diering and Huganir, 2018*). Our data demonstrate that the activation of CPEB3 by neuronal stimulation further facilitates translation of PRPs in vivo. These observations are consistent with a model in which learning induces CPEB3 protein expression, and ablation of CPEB3 abolishes the activity-dependent translation of GluA1 and GluA2 in the mouse hippocampus (*Fioriti et al., 2015*). Specifically, it has been suggested that CPEB3 converts to prion-like aggregates in stimulated synapses that mediate hippocampal synaptic plasticity and facilitate memory storage (*Si and Kandel, 2016*). Because training can produce effective long-term memory, it is likely that increased CPEB3 protein expression due to *Cpeb3* ribozyme inhibition further facilitates experience-induced local translational processes.

ASOs have been used in many studies to inhibit specific mRNAs. A notable example is an FDA-approved ASO that modulates co-transcriptional splicing of the *Smn2* mRNA (*Hua et al., 2010*). More recently, Tran et al. demonstrated that ASO can suppress hexonucleotide repeat expansion of the first intron in the *C9ORF72* gene (*Tran et al., 2022*). Our work shows that an ASO designed to bind the substrate strand of an endogenous self-cleaving ribozyme (located in an intron) increases

the expression of the fully spliced mRNA that harbors the ribozyme. Interestingly, our experiments with inhibitory ASO yielded lower ribozyme levels than control experiments, suggesting that the ASO directs degradation of the target sequence; however, this degradation must occur on a timescale that is longer than the splicing of the mRNA because we consistently measure higher mRNA levels when the ribozyme is inhibited. Given that three endogenous mammalian self-cleaving ribozymes map to introns (*Salehi-Ashtiani et al., 2006*; *de la Peña and García-Robles, 2010*; *Perreault et al., 2011*), we anticipate that application of our ASO strategy will help decipher the effect of these self-cleaving ribozymes on their harboring mRNAs and elucidate their biological roles. Considering ASO as a pharmacological intervention, it is evident that the effect size, as observed in the DI of OLM, is smaller when compared to the *Cpeb3* knockout studies (*Fioriti et al., 2015*). This can be attributed to the mechanisms mediated by the ASO or different training session (10 min vs 15 min), suggesting that the ASO effect has subtle impact on cognitive performance compared to complete genetic ablation.

In summary, our study describes a unique role for the *Cpeb3* ribozyme in post-transcriptional maturation of *Cpeb3* mRNA and its subsequent translation in mouse CA1 hippocampus. Inhibition of the *Cpeb3* ribozyme by ASO and OLM training induces activity-dependent upregulation of CPEB3 and local production of PRPs. These molecular changes are critical for establishing persistent changes in synaptic plasticity that are required for long-term memory. Thus, our study has identified a novel biological role for self-cleaving ribozymes in the brain. More broadly, we have demonstrated a method for determining the biological roles of self-cleaving ribozymes in both mammals (as shown here) and other organisms.

# Materials and methods

**Key resources table**

| Reagent type (species) or resource | Designation | Source or reference | Identifiers | Additional information |
|---|---|---|---|---|
| Strain, strain background (*Mus musculus*) | C57/BL6J | The Jackson Laboratory | Strain # 000664 | |
| Antibody | Anti-CPEB3 (rabbit polyclonal) | Abcam | Cat# ab18833 | 1:1000 |
| Antibody | Anti-GluA1 (mouse monoclonal) | UC Davis/NIH NeuroMab Facility | Cat# 75-327 | 1:1000 |
| Antibody | Anti-GluA2 (rabbit polyclonal) | Proteintech | Cat# 11994-1-AP | 1:2000 |
| Antibody | Anti-PSD-95 (rabbit polyclonal) | Proteintech | Cat# 20665-1-AP | 1:2000 |
| Antibody | Anti-NR2B (rabbit polyclonal) | Proteintech | Cat# 21920-1-AP | 1:2000 |
| Antibody | Anti-GAPDH (mouse monoclonal) | Proteintech | Cat# 60004-1-Ig | 1:10,000 |
| Antibody | Anti-CPEB4 (rabbit polyclonal) | Proteintech | Cat# 25342-1-AP | 1:1000 |
| Antibody | Anti-CPEB1 (rabbit polyclonal) | Abclonal | Cat# A5913 | 1:1000 |
| Antibody | Anti-rabbit HRP (donkey) | Thermo Fisher Scientific | Cat# A16023 | 1:10,000 |
| Antibody | Anti-mouse HRP (goat) | R&D Systems | Cat# HAF007 | 1:1000 |
| Chemical compound, drug | Trizma hydrochloride solution | Sigma-Aldrich | Cat# T2319-1L | |
| Chemical compound, drug | DMSO | VWR | Cat# BDH1115-1LP | |
| Chemical compound, drug | Urea | Sigma-Aldrich | Cat# U5378-5KG | |
| Chemical compound, drug | Acrylamide | Thermo Fisher Scientific | Cat# BP1406-1 | |
| Chemical compound, drug | Triton X-100 | Sigma-Aldrich | Cat# T8787-100ML | |
| Chemical compound, drug | Tris Base | Thermo Fisher Scientific | Cat# BP152-500 | |
| Chemical compound, drug | TWEEN-20 | Sigma-Aldrich | Cat# P9416-100ML | |
| Chemical compound, drug | EDTA | Invitrogen | Cat# 15-575-020 | |
| Chemical compound, drug | [$\alpha$-$^{32}$P]ATP | PerkinElmer | Cat# BLU503H250UC | |

*Continued on next page*

*Continued*

| Reagent type (species) or resource | Designation | Source or reference | Identifiers | Additional information |
|---|---|---|---|---|
| Chemical compound, drug | Neurobasal medium | Thermo Fisher Scientific | Cat# 21103049 | |
| Chemical compound, drug | B27 supplement | Thermo Fisher Scientific | Cat# 17504044 | |
| Chemical compound, drug | Penicillin-streptomycin (10,000 U/mL) | Thermo Fisher Scientific | Cat# 15140122 | |
| Chemical compound, drug | L-Glutamine | Thermo Fisher Scientific | Cat# 25030081 | |
| Chemical compound, drug | Phosphate-buffered saline | Corning | Cat# 21030CV | |
| Chemical compound, drug | TRI reagent | Sigma-Aldrich | Cat# T9424 | |
| Chemical compound, drug | Poly D-lysine | Sigma-Aldrich | Cat# P6407-5MG | |
| Chemical compound, drug | L-Glutamic acid | Sigma-Aldrich | Cat# G1251-100G | |
| Chemical compound, drug | Potassium chloride | Sigma-Aldrich | Cat# P9541-1KG | |
| Chemical compound, drug | Trypan Blue | Corning | Cat# 25-900CI | |
| Chemical compound, drug | GlycoBlue Coprecipitant | Thermo Fisher Scientific | Cat# AM9515 | |
| Chemical compound, drug | 2-Mercaptoethanol | Sigma-Aldrich | Cat# M6250-100ML | |
| Chemical compound, drug | 10× Tris/Glycine/SDS | Bio-Rad | Cat# 1610732 | |
| Chemical compound, drug | 4× Laemmli Sample Buffer | Bio-Rad | Cat# 1610747 | |
| Chemical compound, drug | Restore Western Blot Stripping Buffer | Thermo Fisher Scientific | Cat# PI21059 | |
| Commercial assay or kit | Cell Viability and Proliferation Assays | Biotium | Cat# 30007 | |
| Commercial assay or kit | Pierce BCA Protein Assay Kit | Thermo Fisher Scientific | Cat# 23227 | |
| Commercial assay or kit | SuperSignal West Femto Substrate | Thermo Fisher Scientific | Cat# PI34094 | |
| Commercial assay or kit | RIPA Lysis Buffer System | Santa Cruz Biotechnology | Cat# sc-24948 | |
| Commercial assay or kit | iTaq Universal SYBR Green Supermix | Bio-Rad | Cat# 1725122 | |
| Commercial assay or kit | T7 RNA polymerase | New England Biolabs | Cat# M0251L | |
| Commercial assay or kit | M-MLV Reverse Transcriptase | Promega | Cat# M1701 | |
| Commercial assay or kit | RNase Inhibitor, Murine | New England Biolabs | Cat# M0314S | |
| Commercial assay or kit | DreamTaq PCR Master Mix (2×) | Thermo Fisher Scientific | Cat# K1072 | |
| Software, algorithm | Prism 9 | GraphPad | https://www.graphpad.com/features | |

## Primary cortical neuronal culture

Pregnant female C57BL/6 mice (The Jackson Laboratory) were euthanized at E18 and embryos were collected into an ice-cold Neurobasal medium (Thermo Fisher Scientific). Embryonic cortices were dissected, meninges were removed, and tissues were minced. Cells were mechanically dissociated, passed through a 40 µm cell strainer, counted, and plated at a density of $0.5 \times 10^6$ cells per well in six-well plates coated with poly-D-lysine (Sigma-Aldrich). Neuronal cultures were maintained at 37°C with 5% $CO_2$ and grown in Neurobasal medium containing 2% B27 supplement (Thermo Fisher Scientific), 1% penicillin/streptomycin (Thermo Fisher Scientific), and 2 mM L-glutamine (Thermo Fisher Scientific) for 7–10 days in vitro (DIV), with 50% of the medium being replaced every 3 d. All experimental procedures were performed according to the National Institutes of Health Guide for the Care and Use of Laboratory Animals and approved by the Institutional Animal Care and Use Committee of the University of California, Irvine.

## Mice

C57BL/6J mice (male, 8–10 weeks old, The Jackson Laboratory) were housed in a 12 hr light/dark cycle and had free access to water and food. All experiments were conducted during the light cycle. All experimental procedures were performed according to the National Institutes of Health Guide for the Care and Use of Laboratory Animals and approved by the Institutional Animal Care and Use Committee of the University of California, Irvine (2017-09-14 A3416-01).

## Measurement of co-transcriptional self-scission of the *Cpeb3* ribozyme

In vitro co-transcriptional cleavage kinetics were measured using a previously described method that utilizes standard T7 RNA polymerase in vitro transcription under minimal MgCl$_2$ concentration, followed by a 25-fold dilution of the reaction to stop the synthesis of transcripts and allow the study of the self-scission reaction without the need for purification or additional preparation steps (*Passalacqua et al., 2017*). Transcription reactions were set up in a 5 µL volume and incubated for 10 min at 24°C. The reactions contained the following components: 1 µL of 5× transcription buffer (10 mM spermidine, 50 mM dithiothreitol, 120 mM Tris chloride buffer, pH 7.5, and 0.05% Triton X-100), 1 µL of 5× ribonucleoside triphosphates (final total concentration of 6.8 mM), 1 µL of 5 mM Mg$^{2+}$, 1 µL DNA amplified by PCR to about 1 µM final concentration, 0.5 µL of 100% DMSO, 0.15 µL of water, 0.1 µL of murine RNase inhibitor (40,000 units/mL, New England Biolabs), 0.125 µL of T7 polymerase, and 0.125 µL [α-$^{32}$P]ATP. To prevent initiation of new transcription, the reactions were diluted into 100 µL of physiological-like buffer solution at 37°C. The solution consisted of 2 mM Mg$^{2+}$ (to promote ribozyme self-scission), 140 mM KCl, 10 mM NaCl, and 50 mM Tris chloride buffer (pH 7.5). The 100 µL solution was then held at 37°C for the reminder of the experiment while aliquots were withdrawn at various time points. An equal volume of 4 mM EDTA/7 M urea stopping solution was added to each aliquot collected. Aliquots were resolved using denaturing polyacrylamide gel electrophoresis (PAGE, 7.5% polyacrylamide, 7 M urea). The PAGE gel was exposed to a phosphorimage screen for ~2 hr and analyzed using a Typhoon imaging system (GE Healthcare). Band intensities corresponding to the uncleaved ribozymes and the two products of self-scission were analyzed using ImageQuant (GE Healthcare) and exported into Excel. Fraction intact was calculated as the intensity of the band corresponding to the uncleaved ribozyme divided by the sum of band intensities in a given PAGE lane. The data were fit to a biexponential decay model:

$$k_{\text{obs}} = \text{A} \times \text{e}^{-k(1)t} + \text{B} \times \text{e}^{-k(2)t} + \text{C}$$

In the case of the smallest (minimal) murine *Cpeb3* ribozyme construct (–10/72; *Table 1*), the data were modeled by a monoexponential decay with an uncleaved fraction (using parameters A, $k_1$, and C only).

## In vitro co-transcriptional cleavage kinetics in the presence of ASO

To test inhibition of the *Cpeb3* ribozyme by ASOs, in vitro transcription was performed in a solution containing 10 mM dithiothreitol (DTT), 2 mM spermidine, 4.5 mM MgCl$_2$; GTP, UTP, and CTP (1.25 mM each); 250 µM ATP; 4.5 µCi of [α-$^{32}$P]ATP (PerkinElmer); 40 mM HEPES (pH 7.4), and 1 unit of T7 RNA polymerase. A 5.0 µL transcription reaction was initiated by the addition of 0.5 pmol of DNA template, and the mixture was incubated at 24°C for 10 min. A 1.0 µL aliquot of the reaction was withdrawn, and its transcription and self-scission were terminated by the addition of urea loading buffer. The remaining 4.0 µL volume was diluted 25-fold (final volume of 100 µL) into a physiological-like solution (50 mM HEPES buffer [pH 7.4], 10 mM NaCl, 140 mM KCl, 10 mM MgCl$_2$, and 1 µM of the ASO of interest) at 37°C. A control experiment was performed in the presence of Ctrl-ASO. Then, 5 µL aliquots were collected at the indicated times and terminated by the addition of 5 µL denaturing loading buffer (20 mM EDTA, 8 M urea, and the loading dyes xylene cyanol and bromophenol blue). Samples were resolved on a 10% PAGE under denaturing conditions (7 M urea). The PAGE gel was exposed to a phosphorimage screen and analyzed using Typhoon phosphorimager and ImageQuant software (GE Healthcare). Band intensities were analyzed by creating line profiles of each lane using ImageQuant. Self-cleavage data were fit to a monoexponential decay function:

$$\text{Fraction intact} = \text{A} \times \text{e}^{-kt} + \text{C}$$

where $A$ represents the relative fractions of the ribozyme population cleaving with an apparent rate constant $k$, and $C$ represents the population remaining uncleaved. The model was fit to the data using a linear least-squares analysis and the Solver module of Microsoft Excel.

## Antisense oligonucleotides

ASOs used in this study are 20 nucleotides in length and are chemically modified with 2′-*O*-methoxyethyl (MOE, underlined) and 2′,4′-constrained ethyl (cEt, bold) (*Seth et al., 2009*). All internucleoside linkages are modified with phosphorothioate linkages to improve nuclease resistance. ASOs were solubilized in sterile phosphate-buffered saline (PBS). The sequences of the ASOs are as follows (all cytosine nucleobases are 5-methyl-substituted):

> Scrambled control ASO: 5′-**C**C**T**T**CC**C**T**G**A**AG**G**TT**CCT**CC**-3′;
> *Cpeb3* ribozyme ASO: 5′-**T**GT**G**G**C**C**CCC**TG**T**TA**T**CCT**C**-3′.

## Neuronal stimulation

Neurons were treated with ASO or scrambled ASO (1 µM) for 18 hr prior to neuronal stimulation. To study activity-dependent gene regulation, neuronal cultures were treated with vehicle, 5 µM glutamate (10 min), or 35 mM KCl (5 min). After stimulation, cultures were washed with Hanks' buffered salt solution (HBSS, Thermo Fisher Scientific), and then fresh medium was added.

## Quantitative RT-PCR analysis

Total RNA was isolated from primary cortical neurons or mouse hippocampus using TRI reagent (Sigma-Aldrich) according to the manufacturer's protocol. RNA concentration was measured using a NanoDrop ND-1000 spectrophotometer (Thermo Fisher Scientific). Total RNA was reverse transcribed using random decamers and M-MLV reverse transcriptase (Promega)/Superscript II RNase H reverse transcriptase (Thermo Fisher Scientific). Quantitative RT-PCR was performed on a Bio-Rad CFX Connect system using iTaq Universal SYBR Green Supermix (Bio-Rad). Designed primers were acquired from Integrated DNA Technologies and are provided in *Table 2*. Desired amplicons were verified by melting curve analysis and followed by gel electrophoresis. The starting quantity of DNA from each sample was determined by interpolation of the threshold cycle (CT) from a standard curve of each primer set. Relative gene expression levels were normalized to the endogenous gene *GAPDH*.

## Immunoblotting

Primary cortical neurons or mouse hippocampal tissues were lysed in RIPA lysis buffer with protease inhibitor (Santa Cruz Biotechnology). Crude synaptosomal fractions were prepared as previously described (*Wirths, 2017*). Protein concentrations were measured using bicinchoninic acid (BCA) protein assay (Thermo Fisher Scientific). Protein samples (10–30 µg) were loaded on 10% sodium dodecyl sulfate polyacrylamide (SDS-PAGE) gels and separated by electrophoresis. Gels were electrotransferred onto polyvinylidene fluoride (PVDF) membranes using a semi-dry transfer system (Bio-Rad). Membranes were either blocked with 5% nonfat milk or 5% bovine serum albumin (BSA) in Tris-buffered saline/Tween 20 (0.1% [vol/vol]) (TBST) for 1 hr at room temperature. Membranes were incubated with primary antibodies overnight at 4°C. After primary antibody incubation, membranes were washed three times with TBST and then incubated with secondary antibodies for 1 hr at room temperature. Bands were detected using an enhanced chemiluminescence (ECL) kit (Thermo Fisher Scientific), visualized using Bio-Rad Chemidoc MP imaging system, and analyzed using Image Lab software (Bio-Rad). GAPDH was used as a loading control.

The membranes were initially probed with anti-CPEB3 antibody (Abcam, 1:1000). Following chemiluminescence detection, the membranes were stripped to remove primary and secondary antibodies using a stripping buffer (Thermo Fisher Scientific). Subsequently, the membranes were reprobed with anti-GluA1 antibody (UC Davis/NIH NeuroMab Facility, 1:1000). This process was repeated with two additional rounds of stripping and reprobing, using anti-GluA2 (Proteintech 1:2000) and anti-PSD-95 antibodies (Proteintech 1:2000). After obtaining measurements for all target proteins, the membranes underwent a final round of stripping and reprobing with anti-GAPDH antibody (Proteintech, 1:10,000) to serve as a loading control.

**Table 2.** List of primers used to make ribozyme constructs and measure RNA expression levels.

| Target | | Sequence |
|---|---|---|
| Cpeb3 exons 1-2 | Forward | CTCCCGTTTCCTTCCTCCAG |
| | Reverse | GGGCTGGGTTTTGCTTTTGT |
| Cpeb3 exons 2–3 | Forward | CGATAATGGTAACAATCTGTTGCC |
| | Reverse | CCTTATCATATCCATTAAGGAGTTCTCC |
| Cpeb3 exons 3–6 | Forward | GACCGGAGTAGGCCCTATGA |
| | Reverse | CCAGACGATAAGGCCTGATCA |
| Cpeb3 exons 6–9 | Forward | ACTCTAGAAAGGTGTTTGTTGGAGG |
| | Reverse | TCGAAGGGGTCGTGGAACT |
| Cpeb3 ribozyme cleaved (220 bp amplicon; 18 nts from the cleavage site) | Forward | GTTCACGTCGCGGCC |
| | Reverse | GTGATATAGTGTGTTCTTCAGTGACTCCT |
| Cpeb3 ribozyme uncleaved (283 bp amplicon starting 45 nts upstream and ending 238 nts downstream of the ribozyme cleavage site) | Forward | CCAAGCAGCAGCACAGGTC |
| | Reverse | GTGATATAGTGTGTTCTTCAGTGACTCCT |
| Cpeb3 fourth intron | Forward | CACTCTAGCCTAACTGGTGAGCTC |
| | Reverse | AGTCATTCCAACAGAAATGAAGTACC |
| Gria1 (GluA1) | Forward | GTCCGCCCTGAGAAATCCAG |
| | Reverse | CTCGCCCTTGTCGTACCAC |
| Gria2 (GluA2) | Forward | TGGTACGACAAAGGAGAGTGC |
| | Reverse | ACCAGCATTGCCAAACCAAG |
| Dlg4 (PSD-95) | Forward | TGAGATCAGTCATAGCAGCTACT |
| | Reverse | CTTCCTCCCCTAGCAGGTCC |
| Grin2b (NR2B) | Forward | GCCATGAACGAGACTGACCC |
| | Reverse | GCTTCCTGGTCCGTGTCATC |
| Cpeb1 | Forward | GACTCAGACACGAGTGGCTTCA |
| | Reverse | ACGCCCATCTTTAGAGGGTCTC |
| Cpeb2 | Forward | GAGATCACTGCCAGCTTCCGAA |
| | Reverse | CAATGAGTGCCTGGACTGAGCT |
| Cpeb4 | Forward | TCAGCTCCAGAAGTATGCTCGC |
| | Reverse | GAGTGCATGTCAAACGTCCTGG |
| Gapdh | Forward | TGACCACAGTCCATGCCATC |
| | Reverse | GACGGACACATTGGGGGTAG |

Other antibodies used in the study included anti-NR2B (Proteintech, 1:2000); anti-CPEB1 (ABclonal, 1:1000), CPEB4 (Proteintech, 1:1000); donkey anti-rabbit-HRP (Thermo Fisher Scientific, 1:10,000); and goat anti-mouse-HRP (R&D Systems, 1:1000).

## In vitro XTT cell viability assay

Primary cortical neurons (10,000–20,000 cells/well) were plated onto 96-well plates coated with poly-D-lysine. After 7–14 d, ASOs or scrambled ASOs were added, and the resulting solutions were incubated for 18 hr. Cell viability was determined using the 2,3-bis[2-methoxy-4-nitro-5-sulfophenyl]–2H-tetrazolium-5-carboxyanilide inner salt (XTT) assay according to the manufacturer's protocol (Biotium). The assay utilizes the ability of viable cells with active metabolism to reduce the yellow tetrazolium salt to the soluble orange formazan product using mitochondrial dehydrogenase enzymes. The XTT

reagent was added to each well and incubated for 2–4 hr at 37°C and under 5% $CO_2$. Absorbance was measured at 450 nm with a reference wavelength of 680 nm using a Biotek Synergy HT microplate reader. Results were normalized to control, and all samples were assayed in triplicate.

## Stereotaxic surgeries

C57/BL6J mice (8–10 weeks old, Jackson Laboratory), housed under standard conditions with light-control (12 hr light/12 hr dark cycles), were anaesthetized with an isoflurane (1–3%)/oxygen vapor mixture. Mice were infused bilaterally to the CA1 region of the dorsal hippocampus with ribozyme ASO or scrambled ASO diluted in sterile PBS. The following coordinates were used, relative to bregma: medial-lateral (ML), ± 1.5 mm; anterior-posterior (AP), −2.0 mm; dorsal-ventral (DV), −1.5 mm. ASOs or vehicle (1 nmol/µL) were infused bilaterally at a rate of 0.1 µL/min using a Neuros Hamilton syringe (Hamilton Company) with a syringe pump (Harvard Apparatus). The injectors were left in place for 2 min to allow diffusion, and then were slowly removed at a rate of 0.1 mm per 15 s. The incision site was sutured, and mice were allowed to recover on a warming pad and then were returned to cages. For all surgeries, mice were randomly assigned to the different conditions to avoid grouping same treatment conditions in time.

## OLM tasks

The OLM task was performed to assess hippocampus-dependent memory, as previously described *Vogel-Ciernia and Wood, 2014*. Briefly, naïve C57/BL6J mice (8–12 weeks old; n = 10–12/group; *Cpeb3* ribozyme ASO or scrambled ASO) were trained and tested. Prior to training, mice were handled 1–2 min for 5 d and then habituated to the experimental apparatus for 5 min on six consecutive days in the absence of objects. During training, mice were placed into the apparatus with two identical objects and allowed to explore the objects for 10 min. Twenty-four hours after training, mice were exposed to the same arena, and long-term memory was tested for 5 min, with the two identical objects present, one of which was placed in a novel location. For all experiments, objects and locations were counterbalanced across all groups to reduce bias. Videos of training and testing sessions were analyzed for discrimination index (DI) and total exploration time of objects. The videos were scored by observers blind to the treatment. The exploration time of the objects was scored when the mouse's snout was oriented toward the object within a distance of 1 cm or when the nose was touching the object. The relative exploration time was calculated as a discrimination index (DI = ($t_{novel}$ − $t_{familiar}$) / ($t_{novel}$ + $t_{familiar}$) × 100%). Mice that demonstrated a location or an object preference during the training trial (DI > ± 20) were removed from analysis.

## 3' RACE

Total RNA was extracted from the mouse CA1 hippocampus, and 3' rapid amplification of cDNA ends (3' RACE) was performed to study the alternative polyadenylation. cDNA was synthesized using

**Table 3.** Primers used in 3' RACE.

| Target | Sequence |
| --- | --- |
| 3' RACE adaptor | CCAGTGAGCAGAGTGACGAGGACTCGAGCTCAAGCTTTTTTTTTTTTTTTTTTTTT |
| 3' RACE outer primer | CCAGTGAGCAGAGTGACG |
| 3' RACE inner primer | GAGGACTCGAGCTCAAGC |
| *Gria1* | GGTCCGCCCTGAGAGGTCCC |
| *Gria1* nested | CCTGAGCAATGTGGCAGGCGT |
| *Gria2* | GCTACGGCATCGCCACACCT |
| *Gria2* nested | ATCCTTGTCGGGGGCCTTGGT |
| *Dlg4* | GGCCACGAAGCTGGAGCAGG |
| *Dlg4* nested | GGCCTGGACTCACCCTGCCT |
| *Grin2b* | GAGACGAAGGCTGCAAGCTGGT |
| *Grin2b* nested | CGCCAGGTGGACCTTGCTATCC |

oligo(dT) primers with 3′ RACE adapter primer sequence at the 5′ ends. This cDNA library results in a universal sequence at the 3′ end. A gene-specific primer (GSP) and an anchor primer that targets the poly(A) tail region were employed for the first PCR using the following protocol: 95°C for 3 min, then 30 cycles of 95°C for 30 s, 55°C for 30 s, and 72°C for 3 min, with a final extension of 72°C for 5 min. To improve specificity, a nested PCR was then carried out using primers internal to the first two primers. Upon amplification condition optimization, a quantitative PCR was performed on the first diluted PCR product using the nested primers, and a standard curve of the primer set was generated to measure the relative expression of 3′-mRNA and alternative polyadenylation. All primers used in this study are listed in *Table 3*. When resolved using agarose gel electrophoresis, this nested-primer qPCR produced single bands corresponding to the correct amplicons of individual cDNAs.

## Statistical analysis

Data are presented as means ± SEM. Statistical analyses were performed using GraphPad Prism (GraphPad Prism Software). Statistical differences were determined using (i) two-tailed Welch's *t*-test when comparing between two independent groups, (ii) one-way ANOVA with Šidák's *post hoc* tests when comparing across three or more independent groups, and (iii) two-way ANOVA with Šidák's *post hoc* tests when comparing two factors. $p < 0.05$ was considered significant.

## Acknowledgements

We thank M Malgowska, C-K Lau, and MA Sta Maria for experimental assistance, and L Fioriti and E Kandel for encouragement and support during early parts of the project. This work was supported by the National Institutes of Health grant R01AG051807 (MAW); the National Institutes of Health grant RF1AG057558 (MAW); the National Science Foundation 1804220 (AL); the National Science Foundation 1330606 (AL); the National Science Foundation Graduate Research Fellowship (CCC); and the National Institute of Health grant R01 R01CA229696 (CCC).

## Additional information

### Competing interests

Mehran Nikan: Affiliated with Ionis Pharmaceuticals. The author has no financial interests to declare. The other authors declare that no competing interests exist.

### Funding

| Funder | Grant reference number | Author |
|---|---|---|
| National Institutes of Health | R01AG051807 | Marcelo A Wood |
| National Institutes of Health | RF1AG057558 | Marcelo A Wood |
| National Science Foundation | 1804220 | Andrej Luptak |
| National Science Foundation | 1330606 | Andrej Luptak |
| National Science Foundation | Graduate Research Fellowship Program | Claire C Chen |
| National Institutes of Health | R01CA229696 | Claire C Chen |

The funders had no role in study design, data collection and interpretation, or the decision to submit the work for publication.

### Author contributions

Claire C Chen, Data curation, Formal analysis, Investigation, Visualization, Methodology, Writing – original draft, Project administration, Writing – review and editing; Joseph Han, Carlene A Chinn,

Jacob S Rounds, Marie Myszka, Liqi Tong, Data curation; Xiang Li, Conceptualization; Mehran Nikan, Resources, Writing – review and editing; Luiz FM Passalacqua, Data curation, Formal analysis, Writing – original draft; Timothy Bredy, Conceptualization, Writing – review and editing; Marcelo A Wood, Conceptualization, Supervision, Funding acquisition, Methodology, Writing – review and editing; Andrej Luptak, Conceptualization, Supervision, Writing – original draft, Project administration, Writing – review and editing

### Author ORCIDs
Claire C Chen ⬦ http://orcid.org/0000-0003-2490-2909
Luiz FM Passalacqua ⬦ http://orcid.org/0000-0002-5490-2427
Timothy Bredy ⬦ https://orcid.org/0000-0003-3280-126X
Andrej Luptak ⬦ https://orcid.org/0000-0002-0632-5442

### Ethics
All experimental procedures were performed according to the National Institutes of Health Guide for the Care and Use of Laboratory Animals and approved by the Institutional Animal Care and Use Committee of the University of California, Irvine.

### Decision letter and Author response
Decision letter https://doi.org/10.7554/eLife.90116.sa1
Author response https://doi.org/10.7554/eLife.90116.sa2

---

## Additional files

### Supplementary files
• MDAR checklist

### Data availability
All data generated or analyzed during this study are included in the manuscript and supporting file. Source data files have been provided for Figures 1–7.

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
