## [Editor Report]

In this manuscript the authors describe the expression and regulatory function of a self-cleaving ribozyme in the *Cpeb3* gene. This is an important study because although self-cleaving ribozymes have been identified in the genome, the functions of these RNA enzymes for molecular control for the genes that harbor them is mostly unknown. The manuscript provides compelling data for the molecular function of the ribozyme in gene expression regulation and solid evidence of its role in hippocampal learning. The study will be of interest to neurobiologists who study gene regulatory mechanism.

---

## [Decision Letter]

**Decision letter after peer review:**

Thank you for submitting your article "Inhibition of CPEB3 ribozyme elevates CPEB3 protein expression and polyadenylation of its target mRNAs, and enhances object location memory" for consideration by *eLife*. Your article has been reviewed by 3 peer reviewers, one of whom is a member of our Board of Reviewing Editors, and the evaluation has been overseen by Laura Colgin as the Senior Editor.

Essential revisions:

The reviewers all felt that the topic was important and that going from a ribozyme in an interesting gene to behavior was highly significant. There were, however, several questions raised in the discussion about the details of how the data in Figures 2 and 3 supports the model of ribozyme cleavage and activity. Reviewers 1 and 2 suggested that this might be a result of the text overstating the results and that could be fixed by a clearer description of how ribozyme cleavage and activity is interpreted from the data. However reviewer 3 felt strongly that additional experiments were required to support the model.

Because the measures of ribozyme cleavage and activity are central to the main significance of the study, we did agree that asking for additional experiments would be reasonable in this case if indeed the data provided did not answer these questions. These are the key questions that were raised by all three reviewers and that need to be addressed either by clarification/rewriting or new experiments in the revised version:

1) Does the behavioral paradigm (without the ASO) alter the ribozyme cleavage and intron 3 splicing? The authors should examine whether the tested behavioral paradigm alters the ribozyme activity and intron 3 splicing.

2) Are there data to show that the ASO inhibits cleavage of the ribozyme? Reviewer 3 asks, how were total and uncleaved Ribozymes measured by qRT-PCR? Where are the primers' locations? If the two products were amplified using different primers, their subtraction to derive % cleavage would not be appropriate.

3) What is the evidence for the model of how the ASO works? Reviewer 3 asks, Line 400-403: Shouldn't ribozyme-blocking ASO prevent ribozyme self-cleavage, and as a result should further increase ribozyme levels? This would contradict the result in Figure 3A.

We have included the full reviews from all three reviewers below and encourage you to consider using their comments to guide any additional revisions you might choose to make.

*Reviewer #1 (Recommendations for the authors):*

I have a few suggestions.

1) Page 18, Line 415 says, "The ASO likely prevents CBEP3 ribozyme from cleaving the intron co-transcriptionally and thereby promotes mRNA maturation." I found this sentence a bit confusing because of the "likely" and it made me realize that the authors might need to step the reader through the data needed to truly demonstrate this point. It looks like the ASO is shown in Figure 1F,G to prevent ribozyme cleavage and in Figure 3 the authors do a good job of walking through the expression of different exons. If there is a possible experiment in the middle that would measure the intron cleavage or the speed of mRNA maturation that would be stronger.

2) Ideally to prove that the effects of the ribozyme ASO on expression of PSD95 etc is mediated by CPEB3, the authors would treat CPEB3 knockout/knockdown cells with the ribozyme ASO and show the effect is lost. I do not think this experiment is required but it is what it would take to prove the mechanism so the text should mention this limitation.

3) In a previous review the reviewers had questioned the behavior data on the grounds that the total exploration time was too short to be meaningful. I searched the literature for examples of OLM tasks and found that most (outside the papers reported by UCI researchers) only report the discrimination index without stating the interaction times. However in the papers that did state interaction times I found one from a group not related to this one that used mice and showed interaction times of 5-10sec. By contrast in the papers that reported times and used rats, I found interaction times of 80-100sec. Thus a possible explanation for why the previous reviewer felt these data were not in line with other studies is because that person was accustomed to rat studies. The authors would be served to do this literature search as well and add some context to the OLM studies to this text.

*Reviewer #2 (Recommendations for the authors):*

The CPEB3 ribozyme is an intriguing RNA element that is highly conserved in all mammals, suggesting that it has important functions. The authors developed a very nice anti-sense oligo (ASO) approach to probing the function of the ribozyme by blocking its self-cleavage activity and then observing the molecular and behavioral consequences. The ASOs seem to cause no problems with cultured cells or mice, and therefore the results can reasonably be interpreted as an investigation of the effects of ribozyme activity. I think this is a very important paper in the field, and the strongest evidence to date that ribozymes have important functions in humans and other mammals. The data clearly shows that blocking the ribozyme increases CPEB3 mRNA, CPEB3 protein, the expected downstream products of CPEB3 activity and mouse performance in a memory test. I do have some suggestions for improving the manuscript below, and some important questions to address. The authors should certainly explain all the results of all the controls of the OLM task experiment.

Suggestions

Paragraph starting line 565 – This paragraph does not describe the results of the Ctrl-ASO, as far as I can tell. This result seems important. Did the scrambled ASO result in increased DI or not? Inconclusive data, or contradictory data would not preclude publication, but it should be disclosed.

At Line 362, I feel like this subheading might be too strong given the evidence. There isn't strong evidence that the ribozyme activity increases because of the stimulation. It does appear that the fraction cleaved detected was maximal at 1h in both data sets, which coincides roughly with the max expression (2h for KCl, 1h for Glu). But the subheading as worded suggests causation to me. Could it specifically refer to "correlation" instead, i.e. "ribozyme activity and cpeb3 mRNA expression are correlated post stimulation"? Importantly, the data does not show if the uncleaved fraction is accumulating, or if the cleaved fraction is going away. This latter explanation would be expected if the cleaved RNA is less stable than the uncleaved RNA, and degraded at a faster rate, which would not be surprising. If this were true, the subheading could read "cleaved RNA degradation is accelerated in response to stimulation", but currently, there is no evidence reported in either direction. If the authors relative Ct values could be used to distinguish between these two possibilities, that would be great. If not, it would be nice if the authors could soften the subheading to be more precise about correlation, and add a discussion of possible ways that the ribozyme could be stimulated, that cleaved RNA could be less stable, and other possible explanations to the Discussion section. Or, if my logic is wrong, please correct me!

Paragraph starting Line 392 – I think this first sentence should say "CPEB3 ribozyme activity is correlated with mRNA expression" (which would be a better subheading, above) not "ribozyme expression"? More generally, referring to "ribozyme expression" is a bit problematic, because every mRNA has a ribozyme, so the expression is identical to the mRNA? Maybe "levels of detected ribozyme" would be more accurate, but this would be hard to fit on the figure legends. "Relative [RNA]" could work too.

Also, I understand the hypothesis that blocking ribozyme cleavage could lead to different isoform abundances, but I don't see how the correlation between mRNA expression and cleavage is important for this hypothesis, which is how the paragraph was motivated. I find the link back to the correlation on line 392 at the beginning of this paragraph confusing, and unnecessary. The experiments could be easily motivated because the ribozyme is in an intron, and ribozyme self-cleavage could reasonably be expected to alter splicing of that intron. Minor suggestion – The evidence on modulated splicing seems a little weak, and the authors could consider moving this data later in the manuscript as "evidence towards mechanism", but that is up to the authors about how they would prefer to tell their story.

Decreased ribozyme levels were detected upon ASO treatment in primary neurons and in the hippocampus. A likely explanation would be that blocked ribozyme would lead to more spliced intron that is then degraded or challenging to detect by qPCR, because it would be in lariat form. Could the authors discuss this? The results show more mRNA, more exon-exon junctions, but less ribozyme. I think some possible explanations should be discussed.

The reporting of statistics is a bit hard to follow because there are many multi-way hypothesis tests going on.

The discussion could be improved. I think it contains a little too much introduction and background type material that may not be needed. The section could focus more on interpreting the results, provide possible explanations for results with support from or comparison to the literature.

1) The paragraphs about polyadenylation-induced translation and synaptic strength could be good, but currently there is no clear connection to the results. I think the connection is implied, but could be clearly stated.

2) Compare the anti-correlation between activity and memory found here with the previous memory tests in humans – are they in agreement? If not, discuss.

3) Does fraction cleaved go down after the peak activity because cleaved molecules go down, or because uncleaved molecules accumulate (see above). Discuss these two options and mechanisms for how either of these two things could happen.

4) Compare the effect size of the ASO on OLM task DI compared to other interventions in the literature – is this a big effect or a small effect? It would be good to compare to different classes of intervention, like diet, drugs, gene knockdowns, etc.

5) Consider discussing why the ribozyme may have evolved and persisted in mammals, especially when memory is better when the ribozyme is deactivated by the ASOs? It would be nice to at least acknowledge that this leaves open questions.

*Reviewer #3 (Recommendations for the authors):*

The premise of a comparable timescale between transcription and ribozyme activity as the foundation of the whole thesis was based on in vitro measurement of self-scission half-life and a broadly generalized transcription rate (which actually varies significantly between genes). to strengthen the argument, the authors should provide concrete evidence, for example, determine the cleavage efficiency by altering transcription rates in cells. This can be achieved using CPEB3 mingenes with different promoters and/or polII variants.

The physiological relevance of the proposed mechanism has yet to be demonstrated without ASO interference. For example, does the behavioral paradigm alter the ribozyme activity and intron 3 splicing?

Figure 2B: How were total and uncleaved Ribozymes measured by qRT-PCR? Where are the primers' locations? If the two products were amplified using different primers, their subtraction to derive % cleavage would not be appropriate.

Line 400-403: Shouldn't ribozyme-blocking ASO prevent ribozyme self-cleavage, and as a result should further increase ribozyme levels? This would contradict the result in Figure 3A.

Figure 2B-d: With KCL treatment, the ASO effects on splicing are inconsistent with different primer pairs. This does not support "At an early time point (2 hours post-KCl induction), the ASO-containing culture displayed an increase of spliced mRNA" (line 404-406)

And what is the explanation for the lack of ASO effects on exon 6-9?

Figure 2: The lack of KCL effect under control ASO also appears inconsistent with Figure 2A at 2 hr KCL treatment. Similarly, fig2e-h appears inconsistent with Figure 2A at 24hr KCL treatment. Explanations are needed.

---

## [Author Response]

Essential revisions:The reviewers all felt that the topic was important and that going from a ribozyme in an interesting gene to behavior was highly significant. There were, however, several questions raised in the discussion about the details of how the data in Figures 2 and 3 supports the model of ribozyme cleavage and activity. Reviewers 1 and 2 suggested that this might be a result of the text overstating the results and that could be fixed by a clearer description of how ribozyme cleavage and activity is interpreted from the data. However reviewer 3 felt strongly that additional experiments were required to support the model.Because the measures of ribozyme cleavage and activity are central to the main significance of the study, we did agree that asking for additional experiments would be reasonable in this case if indeed the data provided did not answer these questions. These are the key questions that were raised by all three reviewers and that need to be addressed either by clarification/rewriting or new experiments in the revised version:1) Does the behavioral paradigm (without the ASO) alter the ribozyme cleavage and intron 3 splicing? The authors should examine whether the tested behavioral paradigm alters the ribozyme activity and intron 3 splicing.

We tested whether the CPEB3 ribozyme cleavage and splicing of exons 2 and 3 are regulated by behavioral paradigm using the OLM task. We analyzed the CPEB3 ribozyme expression and ribozyme self-scission in the OLM task and found that training induces CPEB3 expression and ribozyme self-scission is high (see new Figure 6 B–E). The tested behavioral paradigm does not appear to alter the ribozyme activity and intron splicing.

2) Are there data to show that the ASO inhibits cleavage of the ribozyme? Reviewer 3 asks, how were total and uncleaved Ribozymes measured by qRT-PCR? Where are the primers' locations? If the two products were amplified using different primers, their subtraction to derive % cleavage would not be appropriate.

We measured the levels of the total ribozyme by measuring a 220-bp amplicon that starts 18 nts downstream from the ribozyme cleavage site. The uncleaved ribozyme levels were measured using oligos that amplify a region of the intron that starts 45 nts upstream and ends 238 nts downstream of the ribozyme cleavage site. We added this information to the Table of primers in the manuscript. For all PCR oligos we established independent standard curves and calculated RNA levels independently of other amplicons, as noted in the Methods section and now specified in the Results section as well (Page 15). The measurements were thus appropriate for the calculation of the cleaved ribozyme fractions in the various experiments. The fraction ribozyme cleaved was calculated from the uncleaved fraction as the difference between uncleaved fraction and unity (1 – fraction uncleaved), now specified on page 16 of the manuscript. Fraction uncleaved was calculated as [uncleaved ribozyme]/[total ribozyme], as was done previously (see Salehi-Ashtiani et al. Science 313:1788-1792 or Webb et al. Science 326:953).

3) What is the evidence for the model of how the ASO works? Reviewer 3 asks, Line 400-403: Shouldn't ribozyme-blocking ASO prevent ribozyme self-cleavage, and as a result should further increase ribozyme levels? This would contradict the result in Figure 3A.

We showed that the ribozyme is inhibited in vitro (Figure 1F and 1G) and all our data are consistent with ASO inhibition of the ribozyme in cellulo and in vivo. However, we do not have direct evidence for this ribozyme inhibition in vivo, because such an experiment would require a single-molecule FRET-type sensitivity in cells and this assay has not been developed for ribozyme cleavage in cellulo or in vivo. We measured the ribozyme levels by RT-qPCR and observed lower ribozyme levels in presence of ASO in cultured neurons (Figure 3A) as well as in vivo (Figure 5B), which is nominally in contrast to the observations in vitro. However, in these situations we do not measure the co-transcriptional fate of the intron or the ribozyme; rather, we measure the levels of the intron after splicing (evidenced by the increased levels of spliced exons 2–3) when the intron is likely already being degraded. We also do not know what effect the ribozyme ASO has on the intron stability once splicing occurs. Understandably, this is a weakness of the study—and we are fully open about this result— however, given the abundance of evidence that the ribozyme ASO leads to increase of *CPEB3* mRNA under all conditions tested, we feel that there is strong, if indirect, evidence that our model for the ribozyme function is correct. Future studies will examine this issue closer, but a definitive experimental investigation for the mechanism and timing of ribozyme inhibition and intron degradation is out of scope of this study.

Reviewer #1 (Recommendations for the authors):I have a few suggestions.1) Page 18, Line 415 says, "The ASO likely prevents CBEP3 ribozyme from cleaving the intron co-transcriptionally and thereby promotes mRNA maturation." I found this sentence a bit confusing because of the "likely" and it made me realize that the authors might need to step the reader through the data needed to truly demonstrate this point. It looks like the ASO is shown in Figure 1F,G to prevent ribozyme cleavage and in Figure 3 the authors do a good job of walking through the expression of different exons. If there is a possible experiment in the middle that would measure the intron cleavage or the speed of mRNA maturation that would be stronger.

As noted above, we do not have direct evidence for the ribozyme inhibition in cell culture or in vivo. We only have indirect evidence that the ribozyme is inhibited through increased levels of spliced exons, but upon mRNA splicing, introns are rapidly degraded and that appears to be the effect we are observing when inhibiting the ribozyme. As noted above, a direct assay (for example a single-molecule FRET-type experiment) for ribozyme inhibition in vivo has not been developed and a detailed mechanistic study of ribozyme inhibition in live cells is out of scope of this study and will be the topic of future experiments. However, we have analyzed a nuRNA-seq dataset that presents new RNAs expressed after treatment of mice with kainate and found that the kinetics of transcription of CPEB3 are on par with our observations and that the CPEB3 ribozyme self-cleaves cotranscriptionally—that is, there is a gap in the RNA reads at the site of the ribozyme cleavage, whereas reads corresponding to upstream and ribozymes sequences are present one hr after kainate treatment. The data are discussed in the manuscript and presented in Figure 2—figure supplement 1.

2) Ideally to prove that the effects of the ribozyme ASO on expression of PSD95 etc is mediated by CPEB3, the authors would treat CPEB3 knockout/knockdown cells with the ribozyme ASO and show the effect is lost. I do not think this experiment is required but it is what it would take to prove the mechanism so the text should mention this limitation.

We acknowledge the importance of treating CPEB3 knockout/knockdown cells with the ribozyme ASO would indeed provide additional evidence to directly establish the mediating role of CPEB3 and we recognize that the current scope of our study is limited. Our primary focus was to demonstrate the downstream effects of the CPEB3 ribozyme ASO on specific PRPs. However, we agree that addressing the role of CPEB3 in this context would provide a more comprehensive understanding of the underlying mechanism. We plan to address these aspects by incorporating further experiments in future studies.

3) In a previous review the reviewers had questioned the behavior data on the grounds that the total exploration time was too short to be meaningful. I searched the literature for examples of OLM tasks and found that most (outside the papers reported by UCI researchers) only report the discrimination index without stating the interaction times. However in the papers that did state interaction times I found one from a group not related to this one that used mice and showed interaction times of 5-10sec. By contrast in the papers that reported times and used rats, I found interaction times of 80-100sec. Thus a possible explanation for why the previous reviewer felt these data were not in line with other studies is because that person was accustomed to rat studies. The authors would be served to do this literature search as well and add some context to the OLM studies to this text.

We thank the reviewer for their insightful comments on our previous review. We appreciate the effort in searching the literature for examples of OLM tasks and highlighting the variance in reported interaction times between studies in mice and rats. Indeed, during the novel object preference test, it was observed that mice displayed a shorter exploration time and approached the objects at slower pace compared to rats (Stranahan, 2011). We have added introductory sentences and references to the manuscript on page 23 to better describe the task. The interaction times we measured depend on the distance measured, among other variables; that is, the threshold distance of what is considered an interaction. We used 1 cm cutoff and observed interaction times that are typical for this threshold. The details of the measurement are described in the Methods section. Moreover, we have found that the total exploration time depends on the device used to score the OLM data, although the DI scores do not differ by scoring device. Importantly, we avoid scoring mice with too low exploration times by excluding mice from the study if exploration is less than 2 seconds during training or testing. Mice that showed a preference for either object during training (DI > ± 20) were also excluded. Overall, we have demonstrated rigor and reproducibility using this task and the methods to collect and interpret the data across numerous studies, several labs, adult mice and aging mice, and many different genetic and viral manipulations.

Reviewer #2 (Recommendations for the authors):The CPEB3 ribozyme is an intriguing RNA element that is highly conserved in all mammals, suggesting that it has important functions. The authors developed a very nice anti-sense oligo (ASO) approach to probing the function of the ribozyme by blocking its self-cleavage activity and then observing the molecular and behavioral consequences. The ASOs seem to cause no problems with cultured cells or mice, and therefore the results can reasonably be interpreted as an investigation of the effects of ribozyme activity. I think this is a very important paper in the field, and the strongest evidence to date that ribozymes have important functions in humans and other mammals. The data clearly shows that blocking the ribozyme increases CPEB3 mRNA, CPEB3 protein, the expected downstream products of CPEB3 activity and mouse performance in a memory test. I do have some suggestions for improving the manuscript below, and some important questions to address. The authors should certainly explain all the results of all the controls of the OLM task experiment.SuggestionsParagraph starting line 565 – This paragraph does not describe the results of the Ctrl-ASO, as far as I can tell. This result seems important. Did the scrambled ASO result in increased DI or not? Inconclusive data, or contradictory data would not preclude publication, but it should be disclosed.

We have outlined the results of the Ctrl-ASO in the OLM, indicating a notable increase in DI, consistent with previous research. This finding supports the pattern observed in our previous studies (Dong et al., 2022, Keiser et al., 2021, Kwapis et al., 2019, Shu et al., 2018, Kwapis et al., 2018, Vogel-Ciernia et al., 2015), demonstrating a significant increase in DI during the testing session in wild-type mice, compared to the training session.

At Line 362, I feel like this subheading might be too strong given the evidence. There isn't strong evidence that the ribozyme activity increases because of the stimulation. It does appear that the fraction cleaved detected was maximal at 1h in both data sets, which coincides roughly with the max expression (2h for KCl, 1h for Glu). But the subheading as worded suggests causation to me. Could it specifically refer to "correlation" instead, i.e. "ribozyme activity and cpeb3 mRNA expression are correlated post stimulation"? Importantly, the data does not show if the uncleaved fraction is accumulating, or if the cleaved fraction is going away. This latter explanation would be expected if the cleaved RNA is less stable than the uncleaved RNA, and degraded at a faster rate, which would not be surprising. If this were true, the subheading could read "cleaved RNA degradation is accelerated in response to stimulation", but currently, there is no evidence reported in either direction. If the authors relative Ct values could be used to distinguish between these two possibilities, that would be great. If not, it would be nice if the authors could soften the subheading to be more precise about correlation, and add a discussion of possible ways that the ribozyme could be stimulated, that cleaved RNA could be less stable, and other possible explanations to the Discussion section. Or, if my logic is wrong, please correct me!

We agree that the ribozyme expression is correlated with the rest of the *CPEB3* mRNA and we softened the heading to simply state that the *CPEB3* expression is elevated in response to neuronal stimulation. We provide further evidence for the gene expression induction and co-transcriptional ribozyme self-scission by analyzing a published nuRNA-seq dataset (GSE125068) that mapped new RNA reads post neuronal stimulation. As in our experiments and previously published experiments, the number of reads covering early segments of the gene, including the ribozyme increase 1 hour after administration of kainite to mice. These results are in agreement with our observations in primary neurons stimulated by KCl or glutamate, or in mice.

Paragraph starting Line 392 – I think this first sentence should say "CPEB3 ribozyme activity is correlated with mRNA expression" (which would be a better subheading, above) not "ribozyme expression"? More generally, referring to "ribozyme expression" is a bit problematic, because every mRNA has a ribozyme, so the expression is identical to the mRNA? Maybe "levels of detected ribozyme" would be more accurate, but this would be hard to fit on the figure legends. "Relative [RNA]" could work too.Also, I understand the hypothesis that blocking ribozyme cleavage could lead to different isoform abundances, but I don't see how the correlation between mRNA expression and cleavage is important for this hypothesis, which is how the paragraph was motivated. I find the link back to the correlation on line 392 at the beginning of this paragraph confusing, and unnecessary. The experiments could be easily motivated because the ribozyme is in an intron, and ribozyme self-cleavage could reasonably be expected to alter splicing of that intron. Minor suggestion – The evidence on modulated splicing seems a little weak, and the authors could consider moving this data later in the manuscript as "evidence towards mechanism", but that is up to the authors about how they would prefer to tell their story.

We decided to state both that the ribozyme expression and its activity are correlated with the expression of the mRNA, even though half of the statement is rather obvious. We kept the section in its original place in the manuscript.

Decreased ribozyme levels were detected upon ASO treatment in primary neurons and in the hippocampus. A likely explanation would be that blocked ribozyme would lead to more spliced intron that is then degraded or challenging to detect by qPCR, because it would be in lariat form. Could the authors discuss this? The results show more mRNA, more exon-exon junctions, but less ribozyme. I think some possible explanations should be discussed.

As described above, we do not have a way of detecting the ribozyme inhibition directly in vivo. We agree with the reviewer that the spliced-out intron is likely degraded fast, giving the surprising result that the ribozyme ASO leads to lower overall levels of the ribozyme. We have added a statement describing this model on line 420.

The reporting of statistics is a bit hard to follow because there are many multi-way hypothesis tests going on.

There is little we can do about this since we aim to present the statistics with the experiments. We understand that this format makes the Results section somewhat more difficult to read.

The discussion could be improved. I think it contains a little too much introduction and background type material that may not be needed. The section could focus more on interpreting the results, provide possible explanations for results with support from or comparison to the literature.

We chose to keep the Discussion largely intact to bring both the ribozyme and neurobiology fields into perspective. We also added a section on the role of polyadenylation, CPEBs, and local translation in neurobiology. Finally, we added a couple of sentences on the effect of ribozyme ASO, when compared to ablation CPEB3 KO mice.

1) The paragraphs about polyadenylation-induced translation and synaptic strength could be good, but currently there is no clear connection to the results. I think the connection is implied, but could be clearly stated.

We have included a new paragraph in the Discussion section, emphasizing the link between polyadenylation-induced translation and synaptic strength. This addition serves to explicitly establish the connection that underpins and supports our observed results.

2) Compare the anti-correlation between activity and memory found here with the previous memory tests in humans – are they in agreement? If not, discuss.

If the reviewer refers to “activity” as ribozyme activity, i.e. self-scission, that correlation in humans has been suggested based on an association between the episodic memory performance and a SNP in the ribozyme—the faster ribozymes correlate with worse memory in humans. We cite this work in the Introduction.

3) Does fraction cleaved go down after the peak activity because cleaved molecules go down, or because uncleaved molecules accumulate (see above). Discuss these two options and mechanisms for how either of these two things could happen.

As shown in Figure 2, the total ribozyme levels go down over time, whereas the uncleaved population is either steady and low (KCl stimulation) or increases slightly (glutamate stimulation). Most of the effect on apparent ribozyme self-scission is due to decrease in total ribozyme levels, as shown in Figure 2B and 2D. The uncleaved ribozymes are low and are in line with what has been observed in the field of self-cleaving ribozymes in vitro over the past 35 years. At this point we do not have a model for how the levels of uncleaved ribozymes might be modulated.

4) Compare the effect size of the ASO on OLM task DI compared to other interventions in the literature – is this a big effect or a small effect? It would be good to compare to different classes of intervention, like diet, drugs, gene knockdowns, etc.

We have included a discussion on the effect size of ASO on OLM and compared different ASO and other interventions

5) Consider discussing why the ribozyme may have evolved and persisted in mammals, especially when memory is better when the ribozyme is deactivated by the ASOs? It would be nice to at least acknowledge that this leaves open questions.

We chose not to go into this discussion. There are many ways to cleave and destroy an RNA. We imagine that an HDV-like ribozyme originating from a retrotransposon (where they are found in many species) might have landed in the intron of the CPEB3 gene in an early mammal, but we do not know why this functional RNA was retained at such high level of sequence conservation and activity. Future studies correlating activity-based induction of the *CPEB3* gene and the ribozyme self-scission in animals where the ribozyme is either much slower or potentially inactive might shed more light on the role of this catalytic RNA in learning and memory.

Reviewer #3 (Recommendations for the authors):The premise of a comparable timescale between transcription and ribozyme activity as the foundation of the whole thesis was based on in vitro measurement of self-scission half-life and a broadly generalized transcription rate (which actually varies significantly between genes). to strengthen the argument, the authors should provide concrete evidence, for example, determine the cleavage efficiency by altering transcription rates in cells. This can be achieved using CPEB3 mingenes with different promoters and/or polII variants.

The kinetics of transcription of the *CPEB3* gene, as estimated from splicing of its mRNA upon induction, appears roughly to correspond to the speed we cite in the manuscript (Figure 2A and new Figure 6 data). Importantly, we analyzed a dataset of nuRNA-seq performed after administration of kainite to mice (new Figure 2–supplement 1) and find that 1 hr post induction, the reads map to the early segments of the gene, implying the similar kinetics of transcription as discussed in the manuscript and correlating well with the splicing data presented in the manuscript. The nuRNA-seq data show that the ribozyme is synthesized and self-cleaves co-transcriptionally. 6 hrs post-induction all parts of the genes are back to basal levels. Altering transcription rates (and using ribozymes that cleave at different rates) would be an interesting experiment to perform but is out of scope of this work

The physiological relevance of the proposed mechanism has yet to be demonstrated without ASO interference. For example, does the behavioral paradigm alter the ribozyme activity and intron 3 splicing?

We tested whether the CPEB3 ribozyme cleavage and splicing of exons 2 and 3 are regulated by behavioral paradigm using the OLM task. We analyzed the CPEB3 ribozyme expression and ribozyme self-scission in the OLM task and found that training induces CPEB3 expression and ribozyme self-scission is high (see new Figure 6 B–E). The tested behavioral paradigm does not appear to alter the ribozyme activity and intron splicing.

Figure 2B: How were total and uncleaved Ribozymes measured by qRT-PCR? Where are the primers' locations? If the two products were amplified using different primers, their subtraction to derive % cleavage would not be appropriate.

We measured the levels of the total ribozyme by measuring a 220-bp amplicon that starts 18 nts downstream from the ribozyme cleavage site. The uncleaved ribozyme levels were measured using oligos that amplify a region of the intron that starts 45 nts upstream and ends 238 nts downstream of the ribozyme cleavage site. We added this information to the Table of primers in the manuscript. For all PCR oligos we established independent standard curves and calculated RNA levels independently of other amplicons, as noted in the Methods section and now specified in the Results section as well (Page 15). The measurements were thus appropriate for the calculation of the cleaved ribozyme fractions in the various experiments. The fraction ribozyme cleaved was calculated from the uncleaved fraction as the difference between uncleaved fraction and unity (1 – fraction uncleaved), now specified on page 16 of the manuscript. Fraction uncleaved was calculated as [uncleaved ribozyme]/[total ribozyme], as was done previously (see Salehi-Ashtiani et al. Science 313:1788-1792 or Webb et al. Science 326:953).

Line 400-403: Shouldn't ribozyme-blocking ASO prevent ribozyme self-cleavage, and as a result should further increase ribozyme levels? This would contradict the result in Figure 3A.

We showed that the ribozyme is inhibited in vitro (Figure 1F and 1G) and all our data are consistent with ASO inhibition of the ribozyme in cellulo and in vivo. However, we do not have direct evidence for this ribozyme inhibition in vivo, because such an experiment would require a single-molecule FRET-type sensitivity in cells and this assay has not been developed for ribozyme cleavage in cellulo or in vivo. We measured the ribozyme levels by RT-qPCR and observed lower ribozyme levels in presence of ASO in cultured neurons (Figure 3A) as well as in vivo (Figure 5B), which is nominally in contrast to the observations in vitro. However, in these situations we do not measure the co-transcriptional fate of the intron or the ribozyme; rather, we measure the levels of the intron after splicing (evidenced by the increased levels of spliced exons 2–3) when the intron is likely already being degraded. We also do not know what effect the ribozyme ASO has on the intron stability once splicing occurs. Understandably, this is a weakness of the study—and we are fully open about this result— however, given the abundance of evidence that the ribozyme ASO leads to increase of *CPEB3* mRNA under all conditions tested, we feel that there is strong, if indirect, evidence that our model for the ribozyme function is correct. Future studies will examine this issue closer, but a definitive experimental investigation for the mechanism and timing of ribozyme inhibition and intron degradation is out of scope of this study.

Figure 2B-d: With KCL treatment, the ASO effects on splicing are inconsistent with different primer pairs. This does not support "At an early time point (2 hours post-KCl induction), the ASO-containing culture displayed an increase of spliced mRNA" (line 404-406)And what is the explanation for the lack of ASO effects on exon 6-9?

We showed that the ribozyme is inhibited in vitro (Figure 1F and 1G) and all our data are consistent with ASO inhibition of the ribozyme in cellulo and in vivo. However, we do not have direct evidence for this ribozyme inhibition in vivo, because such an experiment would require a single-molecule FRET-type sensitivity in cells and this assay has not been developed for ribozyme cleavage in cellulo or in vivo. We measured the ribozyme levels by RT-qPCR and observed lower ribozyme levels in presence of ASO in cultured neurons (Figure 3A) as well as in vivo (Figure 5B), which is nominally in contrast to the observations in vitro. However, in these situations we do not measure the co-transcriptional fate of the intron or the ribozyme; rather, we measure the levels of the intron after splicing (evidenced by the increased levels of spliced exons 2–3) when the intron is likely already being degraded. We also do not know what effect the ribozyme ASO has on the intron stability once splicing occurs. Understandably, this is a weakness of the study—and we are fully open about this result— however, given the abundance of evidence that the ribozyme ASO leads to increase of *CPEB3* mRNA under all conditions tested, we feel that there is strong, if indirect, evidence that our model for the ribozyme function is correct. Future studies will examine this issue closer, but a definitive experimental investigation for the mechanism and timing of ribozyme inhibition and intron degradation is out of scope of this study.

Figure 2: The lack of KCL effect under control ASO also appears inconsistent with Figure 2A at 2 hr KCL treatment. Similarly, fig2e-h appears inconsistent with Figure 2A at 24hr KCL treatment. Explanations are needed.

The data for KCl are consistent for the early parts of the gene, as discussed above. At 2-hr time point shown in Figure 2A, the exons 2–3 spliced mRNA returns to the basal level, while exons 3–6 are still at higher levels than the starting point. The one segment where there appear to be differences is in the later exons (6–9). One potential explanation for this may be that the samples used for these two experiments came from different mouse cohorts that were not controlled for sex.